# Towards Cold-Start Drafting and Continual Refining: A Value-Driven Memory Approach with Application to NPU Kernel Synthesis

Yujie Zheng [* 1]   Zhuo Li [* 1]   Shengtao Zhang [1]   Jiaqian Wang [1]   Junjie Sheng [2]   Junchi Yan [1]   Weinan Zhang [1]
Ying Wen [1]   Bo Tang [3]   Muning Wen [1]

## Abstract

Deploying Large Language Models to data-scarce programming domains poses significant challenges, particularly for kernel synthesis on emerging Domain-Specific Architectures where a "Data Wall" limits available training data. While models excel on data-rich platforms like CUDA, they suffer catastrophic performance drops on data-scarce ecosystems such as NPU programming. To overcome this cold-start barrier without expensive fine-tuning, we introduce **EvoKernel**, a self-evolving agentic framework that automates the lifecycle of kernel synthesis from initial drafting to continual refining. EvoKernel addresses this by formulating the synthesis process as a memory-based reinforcement learning task. Through a novel value-driven retrieval mechanism, it learns stage-specific Q-values that prioritize experiences based on their contribution to the current objective—whether bootstrapping a feasible draft or iteratively refining latency. Furthermore, by enabling cross-task memory sharing, the agent generalizes insights from simple to complex operators. By building an NPU variant of KernelBench and evaluating on it, EvoKernel improves frontier models' correctness from 11.0% to 83.0% and achieves a median speedup of $3.60\times$ over initial drafts through iterative refinement. This demonstrates that value-driven experience accumulation allows general-purpose models to master the kernel synthesis task on niche hardware ecosystems.

*Equal contribution  [1]Shanghai Jiao Tong University, Shanghai, China [2]Independent Researcher [3]MemTensor (Shanghai) Technology Co., Ltd, Shanghai, China. Correspondence to: Muning Wen <muningwen@sjtu.edu.cn>.

*Proceedings of the $43^{rd}$ International Conference on Machine Learning*, Seoul, South Korea. PMLR 306, 2026. Copyright 2026 by the author(s).

## 1. Introduction

A practical limitation when deploying Large Language Models (LLMs) to niche domains is their inability to generalize beyond their pre-training distribution (Minaee et al., 2024; Wang et al., 2025). When faced with *cold-start* scenarios, domains where training data is sparse and expert demonstrations are unavailable, even frontier models struggle significantly (Kostikova et al., 2025; Joel et al., 2024). This challenge is particularly acute in domains where (i) correctness is binary, leaving little room for "partially correct" solutions (Jain et al., 2025; Yan et al., 2024), (ii) expert knowledge is scarce and expensive to acquire, and (iii) the gap between in-distribution and out-of-distribution performance is stark.

Automated kernel synthesis for emerging hardware accelerators exemplifies this extreme scarcity (Yu et al., 2026). While the industry is aggressively diversifying toward Domain-Specific Architectures (DSAs) like NPUs, TPUs, and neuromorphic chips (Silvano et al., 2025; Liao et al., 2021; Jouppi et al., 2023) to address escalating computational costs (Kaplan et al., 2020), these nascent ecosystems face a severe "Data Wall". Unlike the mature NVIDIA landscape, where decades of CUDA repositories provide a massive pre-training corpus, emerging platforms are characterized by extreme data scarcity: public code is rare, documentation is esoteric, and compiler feedback is opaque (Joel et al., 2024). This barrier is compounded by the fact that highly optimized CUDA kernels (Choquette et al., 2021; Ansel et al., 2024) are not portable to these architectures due to fundamental differences in memory hierarchy and instruction sets, leaving foundation models with virtually no expert demonstrations to bridge the cold-start gap.

As evidenced in Table 1, state-of-the-art LLMs that achieve high performance on CUDA (Ouyang et al., 2025) suffer a catastrophic collapse when transferred to a data-scarce Domain-Specific Language (DSL) like Ascend C, which is specifically designed for NPU kernel programming. In line with prior findings (Wen et al., 2025), even GPT-5.2, which attains 92% on CUDA L1 tasks, drops to 14% on Ascend C; on the more challenging L2 tasks, models fail entirely. This observation suggests that current models do not genuinely

*Table 1.* Few-shot functional correctness (pass@4) of frontier LLMs on CUDA vs. Ascend C kernel generation. Results are from our experiments, with level definitions (L1, L2) and setup details consistent with Section 4.1.

| Model | Level | CUDA (%) | Ascend C (%) |
|---|---|---|---|
| GPT-5.2 | L1 | 92.0 | 14.0 |
| | L2 | 90.0 | 2.0 |
| DeepSeek-V3.2 | L1 | 50.0 | 8.0 |
| | L2 | 9.0 | 0.0 |
| Qwen3-Coder-30B | L1 | 46.0 | 7.0 |
| | L2 | 10.0 | 0.0 |

"learn" to program new hardware like NPUs, but instead rely on memorized patterns from pre-training distributions.

Standard paradigms to bridge this gap prove insufficient in such data-scarce domains. Supervised Fine-Tuning (SFT) (Zhou et al., 2023; Chung et al., 2024) demands thousands of expert-labeled examples per domain (Longpre et al., 2023), which is prohibitively expensive when targeting rapidly evolving or niche environments like NPU programming. Parametric policy-based Reinforcement Learning (Zhang et al., 2025; Kakade, 2003) requires extensive online rollouts to update model weights, incurring high sample complexity (Cao et al., 2024; Qi et al., 2025) and risking catastrophic forgetting of general capabilities. Traditional Retrieval-Augmented Generation (RAG) (Lewis et al., 2020) falters when the database is sparse (Contal & McGoldrick, 2024; Barnett et al., 2024), and even with relevant samples, similarity-based retrieval does not guarantee effectiveness (Izacard et al., 2023). Consequently, the core challenge is a **cold-start** problem: *How can an agent autonomously master a rigorous, data-scarce kernel synthesis task from scratch, without expert demonstrations or expensive fine-tuning?*

To address this, we introduce **EvoKernel**, a framework that formulates kernel synthesis as a reinforcement learning task over a self-evolving memory. By employing a novel value-driven retrieval mechanism, the agent learns stage-specific Q-values to quantify the utility of historical experiences, dynamically shifting focus from bootstrapping functional correctness (Drafting) to optimizing latency (Refining) without updating model weights. Empirically, EvoKernel bridges the cold-start gap on NPU benchmarks, boosting the correctness of frontier models *from 11.0% to 83.0%* and achieving a *3.60x median speedup* over the first feasible draft, thereby demonstrating that value-driven experience accumulation enables general-purpose models to master data-scarce hardware ecosystems.

Our contributions are summarized as follows:

- **Unified Drafting-Refining Pipeline:** We propose a two-stage framework over a shared memory that transitions from feasibility-driven drafting to latency-driven refining to bootstrap and optimize NPU kernels.
- **Evolving Value-Driven Retrieval:** We introduce a retrieval mechanism that learns stage-specific Q-values to quantify memory utility. A unified Monte-Carlo update adapts the policy from verifier feedback without updating model weights.
- **Comprehensive Evaluation and Insights:** EvoKernel boosts performance on NPU benchmarks from 11.0% to 83.0%. We provide in-depth analysis of cross-task transfer and emergent curricula, demonstrating how memory autonomously bridges the data-scarce gap.

## 2. Related Work

**Self-Evolving and Adaptive Agents.** While Large Language Models (LLMs) are typically deployed with fixed parameters, recent work explores mechanisms for self-improvement. Inference-time methods such as Self-Refine (Madaan et al., 2023) and Tree-of-Thoughts (Yao et al., 2023) improve reasoning through iterative refinement, self-evaluation, or search within a single episode. Reflexion (Shinn et al., 2023) extends this idea by storing verbal reflections in task-local episodic memory and reinjecting them into later prompts, enabling within-task cross-trial improvement without weight updates. Closest to our work are evolutionary frameworks like AlphaEvolve (Novikov et al., 2025) and EvolveR (Wu et al., 2025), which accumulate experience across episodes. However, these methods implicitly assume past attempts can be critiqued, ranked, or abstracted into reusable guidance. In data-scarce kernel synthesis, the utility of past experience is weakly correlated with textual or semantic similarity, so we learn value-driven retrieval from verifier outcomes rather than from surface features or textual reflection.

**Memory-Augmented Generation.** To overcome context limitations, systems such as MemGPT (Packer et al., 2023), MemOS (Li et al., 2025c), and Mem0 (Chhikara et al., 2025) introduce operating-system-like memory hierarchies for long-horizon tasks. Agentic systems further show that stored experience can improve future behavior: Voyager (Wang et al., 2024) retrieves executable skills, Memp (Fang et al., 2025) distills procedural memory, and Generative Agents and MemPrompt store behavioral reflections or user-feedback cases (Park et al., 2023; Madaan et al., 2022). More recently, Memento (Zhou et al., 2025) and MemRL (Zhang et al., 2026) learn retrieval policies from feedback rather than relying on passive semantic matching. We adapt this value-based retrieval paradigm to cold-start kernel engineering by coupling a pre-seeded, hybrid memory with stage-specific value estimates, addressing settings where surface-level similarity is a poor signal of utility and purely self-bootstrapped memory cannot get off the ground.

**Automated Kernel Synthesis.** Kernel synthesis demands strict functional correctness and hardware-specific optimization. Benchmarks such as KernelBench (Ouyang et al., 2025) show that frontier LLMs can generate correct and performant CUDA kernels for a subset of tasks, while MultiKernelBench (Wen et al., 2025) and Triton-Bench (Li et al., 2025a) reveal a broader generalization gap across less-represented hardware platforms and specialized DSLs. Recent agentic kernel-optimization frameworks such as QiMeng-Kernel (Zhu et al., 2026), KernelBand (Ran et al., 2026), STARK (Dong et al., 2025), and AKG Kernel Agent (Du et al., 2025) leverage execution feedback, profiling, search, or agent coordination to refine candidate kernels. A parallel line of training-based work, including Kevin (Baronio et al., 2025), AutoTriton (Li et al., 2025b), TritonRL (Woo et al., 2025), and AscendKernelGen (Cao et al., 2026), improves kernel generation through domain-specific fine-tuning and reinforcement learning. Prior kernel agents mainly use feedback to refine the current search or train the generator, which is brittle when cold-start backends provide few viable candidates. EvoKernel instead uses seeded domain memory to bootstrap the first correct kernels and verifier-derived values to identify which experiences should be reused for later refinement.

## 3. EvoKernel: Value-Driven Memory Update for Kernel Evolution

As shown in Figure 1, we propose the EvoKernel, a framework that automates the lifecycle of NPU kernel synthesis, from cold-start drafting to continual performance refinement. We formulate this process as a Memory-based Markov Decision Process (M-MDP) (Zhou et al., 2025; Zhang et al., 2026), where an agent learns to retrieve high-utility experiences to guide a LLM generator.

### 3.1. Problem Formulation

A kernel synthesis task $x \in \mathcal{X}$ is specified by a PyTorch reference operator and metadata (e.g., input shapes and operator hyperparameters). Given a task $x$ and retrieved context $c$, a generator $G_\theta$ samples a kernel and the goal is to generate a kernel source code $y \in \mathcal{Y}$ that satisfies functional correctness and minimizes execution latency.

We model the generation process as an M-MDP over a horizon $T$. A trajectory is defined as $\tau = (s_0, c_0, a_0, r_0, \ldots, s_T)$, governed by the tuple $(\mathcal{S}, \mathcal{A}, \mathcal{M}, \mathcal{P}, \mathcal{R})$. The components are defined as follows:

**State Space ($\mathcal{S}$):** A state $s_t$ is defined as a tuple $(x, \xi_t)$, where $x \in \mathcal{X}$ denotes the static kernel task (PyTorch operator + metadata), and $\xi_t$ represents the *dynamic generation state* (e.g., current best-so-far latency or verification status).

**Action Space ($\mathcal{A}$):** The action $a_t \in \mathcal{A}$ corresponds to a generated kernel code $y \in \mathcal{Y}$.

**Memory ($\mathcal{M}$):** We define $\mathcal{M}_t$ as a dynamic, self-evolving memory bank. It is initialized as $\mathcal{M}_0$ comprising seed knowledge. At each step $t$, it accumulates the agent's interaction history, updating according to the rule:

$$\mathcal{M}_{t+1} \leftarrow \mathcal{M}_t \cup \{(s_t, a_t, r_t)\}, \tag{1}$$

**Transition Dynamics ($\mathcal{P}$):** The transition dynamics $\mathcal{P} : \mathcal{S} \times \mathcal{A} \rightarrow \Delta(\mathcal{S})$ describe the evolution of the generation process. Since task $x$ remains invariant within an episode, $\mathcal{P}$ deterministically updates the generation state:

$$s_{t+1} = (x, \xi_{t+1}), \quad \xi_{t+1} = f(x, \xi_t, a_t, o_t), \tag{2}$$

Here, $f$ updates the dynamic generation state by integrating the action $a_t$ and its verifier outcome $o_t$, conditioned on the task $x$ and the previous state $\xi_t$.

**Reward Function ($\mathcal{R}$):** The environment provides a scalar feedback signal $r_t \in \mathbb{R}$ based on evaluation of the action $a_t$.

**Policy Factorization.** To tackle this M-MDP, the agent operates via a composite policy. At each step $t$, a Retrieval Policy $\mu$ first selects a context $c_t \subset \mathcal{M}_t$ based on the current state. Conditioned on this context, the Generator Policy $G_\theta$ samples the code:

$$\pi(a_t|s_t, \mathcal{M}_t) = G_\theta(a_t|s_t, c_t) \cdot \mu(c_t|s_t, \mathcal{M}_t), \tag{3}$$

Our core methodology focuses on optimizing $\mu$ via reinforcement learning to identify high-utility memory items, while $G_\theta$ leverages the pre-trained capabilities of the LLM.

### 3.2. Memory Architecture and Value-Driven Retrieval

The efficacy of the generator $G_\theta$ depends critically on the quality of the context $c_t$. We design $\mathcal{M}$ as a heterogeneous knowledge base containing: (i) API templates for the target architecture (Ascend C), (ii) best practices for kernel refinement, (iii) summarized success and failure experiences, and (iv) generation traces, including both draft and refined variants. Items (i) and (ii) are pre-seeded as $\mathcal{M}_0$ from documentation and expert summaries, while (iii) and (iv) are accumulated online from verifier feedback.

To instantiate the policy $\mu$, we introduce **Value-Driven Retrieval**. Unlike traditional similarity-based retrieval, our approach dynamically evaluates memory item utility based on the current generation stage. For state $s$ and candidate memory item $m$, we define a Q-value function $Q_k(s, m)$ that estimates the expected benefit of including $m$ in the context at stage $k$.

For a given task $x$, we begin by retrieving a pool of candidates $\mathcal{C}(x) \subset \mathcal{M}$ based on dense retrieval, filtering for semantic relevance. To guide the final context selection, we learn stage-specific value estimates $Q_k$, which reflect the agent's evolving objectives:

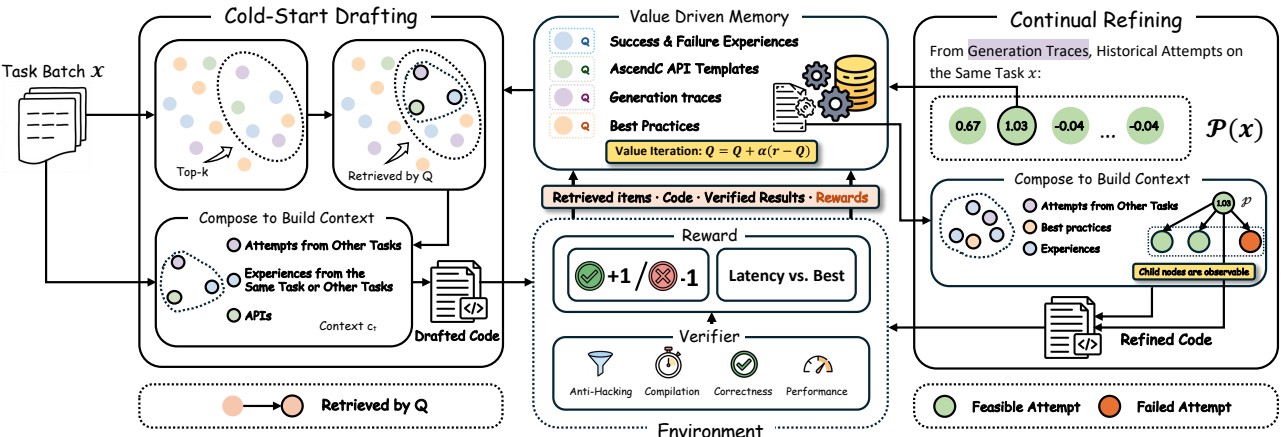

*Figure 1.* The EvoKernel framework. **(Left) Cold-Start Drafting:** Given task batch $\mathcal{X}$, retrieves top-$k$ candidates, filters context via $Q$, and synthesizes an initial kernel. **(Center) Environment & Memory:** A multi-gate verifier assesses generated code to yield rewards, which update $Q$ via value iteration, and the code together with verifier results are stored in Memory. **(Right) Continual Refining:** Exploits generation traces $\mathcal{P}(x)$ and historical attempts, including observable child nodes, to iteratively optimize for lower latency.

- **Drafting Stage ($Q_1$):** Estimates the likelihood that $m$ contributes to a *functionally correct* kernel.
- **Refining Stage ($Q_2$):** Estimates the contribution of a memory item $m$ to *latency optimization* of the kernel, where $m$ can either be an optimization start point $p$ or auxiliary refinement items from $\mathcal{M}$.

**Unified Value Update.** Despite the distinct objectives, we employ a unified Monte-Carlo (MC) update rule to refine the retrieval policy $\mu$. Upon observing a reward $r_t$ (defined in subsequent sections) after using context items $c_t$, we update the Q-values for all $m \in c_t$:

$$Q(s, m) \leftarrow Q(s, m) + \alpha \cdot \big( r_t - Q(s, m) \big), \quad (4)$$

where $\alpha$ is the step size. This update rule allows the retrieval policy $\mu$ to continuously adapt to the evolving capabilities of $G_\theta$. We provide formal guarantees on boundedness and convergence of these value estimates in Appendix A.

Under this update rule, items that contribute to successful generations have their Q-values increased and are retrieved more frequently, while those that do not have their Q-values decreased and are gradually filtered out from future retrieval.

### 3.3. Stage 1: Cold-Start Drafting

The objective of this stage is to obtain an initial *feasible* kernel that can bootstrap subsequent refinement. For a task $x$, we iteratively (i) retrieve a drafting context $c_t \subset \mathcal{C}(x)$ using an $\epsilon$-greedy policy over $Q_1$, and (ii) sample a candidate kernel $y_t \sim G_\theta(\cdot \mid x, c_t)$.

**Reward and update.** We use a binary feasibility reward rather than softer shaping such as partial compilation progress that could credit code appearing promising but ultimately failing end-to-end verification, since correctness is

a prerequisite for any latency optimization:

$$r_{1,t} = \begin{cases} +1, & \text{if } g_{\text{feas}}(o_t) = 1, \\ -1, & \text{otherwise}, \end{cases} \quad (5)$$

where $o_t = V(x, y_t)$ and $g_{\text{feas}}$ is the combined feasibility gate (Section 3.5). After receiving feedback, we update the values of retrieved entries $m \in c_t$ using Eq. 4 with $r = r_{1,t}$ and store the generated code together with verifier feedback into memory. This process repeats until a feasible kernel is found or the budget is exhausted.

### 3.4. Stage 2: Continual Refining

Once a feasible kernel is obtained, the focus shifts from feasibility to *latency reduction*. We maintain a set of *optimization start points* $\mathcal{P}(x)$, initialized with the successful draft from Stage 1 and augmented online as new feasible variants are discovered. At each iteration, based on the current state $s_t$, we retrieve the available start points from the memory $\mathcal{M}$ and select a start point using $Q_2$.

With the selected start point and the current state, we then retrieve additional contextual information that contains optimization traces, best practices, and information about its observable child nodes to support the refinement process. Using the selected start point and the retrieved context in $c_t$, the generator samples a refined result.

**Relative reward, normalization, and update.** To make refinement gains comparable across operators with very different runtime scales, we drive performance optimization by a bounded log-ratio against the best-so-far latency $b_t$ tracked in $\xi_t$, with $\tanh$ keeping the reward symmetric

*Table 2.* Compilation Rate (CR) and Correctness (Acc) across difficulty levels, shown as **(Round 1) Final**, respectively representing the start point and the final performance. Notably, the huge gap between GPT-5.2 and other models implies that frontier LLMs with stronger in-context learning capability benefit substantially more from experience-driven methods.

| Model | Method | Level 1 | | Level 2 | | Overall | |
|---|---|---|---|---|---|---|---|
| | | CR (%) | Acc (%) | CR (%) | Acc (%) | CR (%) | Acc (%) |
| Qwen3-Coder-30B | Pass@$k$ | (22.0) 30.0 | (7.0) 8.0 | (0.0) 2.0 | (0.0) 0.0 | (11.0) 16.0 | (3.5) 4.0 |
| | Refinement | (13.0) 22.0 | (2.0) 6.0 | (0.0) 1.0 | (0.0) 0.0 | (6.5) 11.5 | (1.0) 3.0 |
| | **Ours** | **(25.0) 33.0** | **(6.0) 11.0** | **(1.0) 3.0** | **(0.0) 0.0** | **(13.0) 18.0** | **(3.0) 5.5** |
| DeepSeek-V3.2 | Pass@$k$ | (21.0) 33.0 | (7.0) 9.0 | (1.0) 13.0 | (0.0) 0.0 | (11.0) 23.0 | (3.5) 4.5 |
| | Refinement | **(16.0) 44.0** | (0.0) 12.0 | **(2.0) 26.0** | (0.0) 0.0 | **(9.0) 35.0** | (0.0) 6.0 |
| | **Ours** | (9.0) 39.0 | **(2.0) 19.0** | (1.0) 19.0 | (0.0) 0.0 | (5.0) 29.0 | **(1.0) 9.5** |
| GPT-5.2 | Pass@$k$ | (24.0) 36.0 | (9.0) 19.0 | (2.0) 13.0 | (1.0) 3.0 | (13.0) 24.5 | (5.0) 11.0 |
| | Refinement | (19.0) 88.0 | (7.0) 41.0 | (2.0) 55.0 | (1.0) 3.0 | (10.5) 71.5 | (4.0) 22.0 |
| | Codex | (34.0) 82.0 | (16.0) 70.0 | (16.0) 84.0 | (0.0) 22.0 | (25.0) 83.0 | (8.0) 46.0 |
| | **Ours** | **(20.0) 97.0** | **(7.0) 90.0** | **(2.0) 100.0** | **(1.0) 76.0** | **(11.0) 98.5** | **(4.0) 83.0** |

across improvements and regressions:

$$r_{2,t} = \begin{cases} -1, & \text{if } g_{\text{feas}}(o_t) = 0, \\ \tanh(\log b_t - \log \ell_{\text{lat}}(o_t)), & \text{otherwise.} \end{cases}$$
(6)

We further apply online z-score normalization $\hat{r}_{2,t} = (r_{2,t} - \mu_2)/\sigma_2$ with running estimates $(\mu_2, \sigma_2)$ to keep Q-value updates on a consistent scale as the typical magnitude of $r_{2,t}$ shrinks with $b_t$ improving. We update $Q_2$ for both the start point $p_t$ and retrieved entities $z \in c_t$ using Eq. 4 with $r = \hat{r}_{2,t}$. When a refined kernel is feasible, as indicated by $g_{\text{feas}}(o_t) = 1$, we store the kernel together with verifier feedback in memory for future retrieval and add it to the start set $\mathcal{P}(x)$ to expand the refinement search space.

### 3.5. Multi-gate Verification

The verifier $V$ acts as the environment interface, providing robust feedback to guide the RL process. Given a task $x$ and a generated kernel $y_t$, it returns a structured outcome

$$o_t = V(x, y_t) = (g_{\text{hack}}, g_{\text{comp}}, g_{\text{corr}}, \ell_{\text{lat}}), \quad (7)$$

where $g_{\text{hack}}, g_{\text{comp}}, g_{\text{corr}} \in \{0, 1\}$ denote the anti-hacking, compilation, and correctness gates, and $\ell_{\text{lat}} \in \mathbb{R}_+$ is the measured latency. A kernel is deemed feasible if and only if: $g_{\text{feas}}(o_t) \triangleq g_{\text{hack}} \wedge g_{\text{comp}} \wedge g_{\text{corr}}$.

**Anti-hacking ($g_{\text{hack}}$).** We implement a two-tier screening process. A rule-based filter first rejects trivial exploits (e.g., using high-level `torch` APIs or constant-folding shortcuts). Survivors undergo a model-based inspection to identify subtle harness manipulations.

**Compilation ($g_{\text{comp}}$) & Correctness ($g_{\text{corr}}$).** We verify successful compilation under the Ascend C toolchain. Correctness is validated by comparing outputs against the PyTorch reference: $\|\text{out}_y(x) - \text{ref}(x)\| \leq \tau$. The verifier provides

fine-grained feedback, including mismatch localization and shape errors (details in Appendix G).

**Latency ($\ell_{\text{lat}}$).** For feasible kernels, we measure on-device execution time using the native **msprof** tool. We report the mean wall time across 3 profiling passes (Pipe, Memory, Resource) after warm-up.

## 4. Experiment

### 4.1. Experimental Setup

**Benchmark and Execution.** We evaluate on L1 and L2 operators from KernelBench (Ouyang et al., 2025). Since KernelBench does not natively support Ascend C, we implement a compilation, deployment, and execution pipeline that maintains full compatibility with KernelBench PyTorch references while enabling the model to generate complete Ascend operator projects.

**Budget and metric.** We enforce a strict per-operator budget of $T = 30$ iterations across all methods, encompassing both draft generation and iterative refinement. Functional correctness is verified with tolerances of `atol = rtol =` $10^{-2}$. Our evaluation relies on three primary metrics: (i) **Compilation Rate (CR)**, which measures the proportion of generated kernels that successfully compile; (ii) **Correctness (Acc)**, which reports the percentage of operators for which a functionally valid solution is found within the budget; and (iii) **Speedup**, which measures the reduction in execution latency, defined as speedup $= L_{\text{ref}}/L_{\text{opt}}$, where $L_{\text{ref}}$ and $L_{\text{opt}}$ are the latencies of the reference and optimized kernels, respectively.

**Baselines.** We compare EvoKernel against three baseline strategies using three models: Qwen3-Coder-30B-A3B-Instruct (Yang et al., 2025), DeepSeek-V3.2 (Liu et al., 2025), and GPT-5.2. Detailed configurations of these base-

lines can be found in Appendix D.

- *Pass@k*: A stateless baseline generating $K = 30$ independent candidates per operator given a single demonstration.
- *Refinement*: A stateful agentic loop that iteratively repairs compilation and correctness errors using verifier feedback. Upon finding a valid kernel, it transitions to hill-climbing for latency optimization, subject to a maximum budget of 30 iterations.
- *Codex by OpenAI*: An autonomous agent based on GPT-5.2 with direct shell and file system access. It executes a "try-fail-evolve" loop, autonomously mutating the implementation based on execution logs until success or a budget of 30 verification attempts is exhausted.

### 4.2. Main Results

We evaluate EvoKernel under a matched evaluation pipeline, focusing on compilation and correctness, as well as performance optimization after correctness.

**Compilation and correctness.** Table 2 reports compilation rate (CR) and correctness (Acc) across two difficulty levels under a fixed budget $T=30$.

EvoKernel achieves the strongest overall performance with GPT-5.2, reaching **98.5%** CR and **83.0%** Acc, substantially outperforming Codex (83.0% CR, 46.0% Acc) and Refinement (71.5% CR, 22.0% Acc). On Level 2, EvoKernel attains near-perfect compilation (100%) with 76% correctness. Despite Codex having autonomous shell and file system access, EvoKernel surpasses it by 15.5 points in CR and 37.0 points in Acc.

On weaker backbones, the improvements are more moderate. EvoKernel achieves the highest Acc on both Qwen3-Coder-30B (5.5% vs. 4.0%) and DeepSeek-V3.2 (9.5% vs. 6.0%), with DeepSeek-V3.2 reaching 19% correctness on Level 1—more than doubling Pass@k. The Refinement baseline attains higher CR on DeepSeek-V3.2 (35.0% vs. 29.0%), suggesting that value-driven retrieval prioritizes generation quality over compilation attempts. Critically, Level 2 Acc remains at 0% for weaker models even when candidates compile (e.g., 19% CR on DeepSeek-V3.2), indicating that harder operators demand stronger generator capacity.

Examining Round 1 through the final iteration reveals how effectively each method leverages the iterative process. On GPT-5.2, EvoKernel improves CR from 11.0% to 98.5% and Acc from 4.0% to 83.0%, representing an order-of-magnitude gain. In contrast, weaker models show limited improvement: Qwen3-Coder-30B increases Acc by +2.5 points, while DeepSeek-V3.2 improves by +8.5 points. This disparity reveals a key insight: the in-context learning capabilities of frontier LLMs prove critical for experience-driven approaches like ours. Crucially, it does not weaken our method's value. Instead, it confirms that our agent is keeping pace with the cutting-edge advancements of base models.

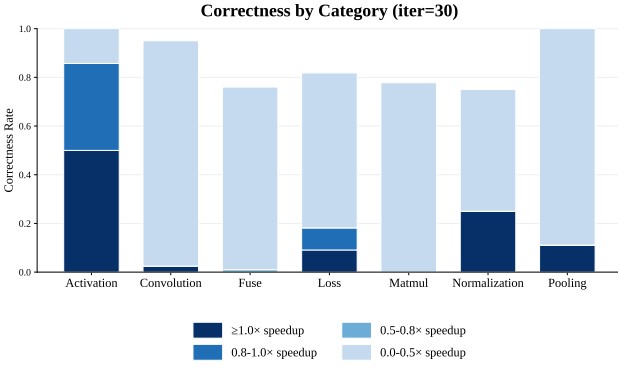

*Figure 2.* Category-level correctness and speedup distribution at budget $T=30$. Color segments show the fraction of correct kernels in each speedup tier relative to Torch-NPU.

**Optimization gains: within-operator speedup.** Conditioned on reaching a correct draft, the refining stage further reduces latency. For each solved operator, we compare the initial draft, defined as the *first feasible* candidate, to the *best* candidate found within the remaining budget. This yields a median speedup of **3.60**×, with an interquartile range of **1.38–10.05**×, and a per-category breakdown is provided in Appendix B. Across the 166 correct operators, 12 match or outperform Torch-NPU. Although many operators remain slower than Torch-NPU (Figure 2), consistent within-operator gains indicate that the refinement process continues to improve performance beyond correctness.

Figure 3 quantifies these gains across the 166 correct operators, and Figure 4 zooms in on four representative ones. The distribution is long-tailed: while many operators exhibit modest improvements ($s \approx 1$–2×), a substantial subset benefits dramatically from continued optimization, with top performers achieving more than $200\times$ speedup over their first correct version.

Iteration-level trajectories for these four representative operators (Figure 4) confirm that the gains in Figure 3 emerge from systematic, incremental improvements across multiple iterations, rather than from single fortuitous generations.

### 4.3. Generalization of Value-Driven Memory

A core motivation for our memory design is *reusability*: high-utility past experiences should accelerate learning on subsequent ones. We verify this hypothesis by evaluating transfer across difficulty levels and generator backbones.

**Transfer across difficulty levels.** We study whether memory accumulated on easier L1 operators transfers to harder

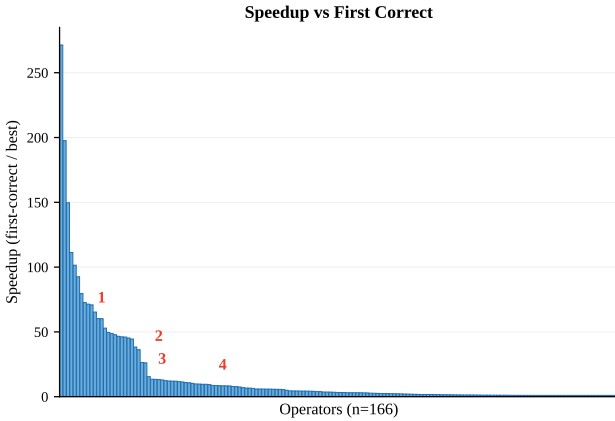

*Figure 3.* Within-operator speedup achieved by iterative refinement, across the 166 correct operators. Each bar is the per-operator ratio (first-correct latency / best latency), sorted in descending order. Bars annotated 1–4 mark the four representative operators whose iteration-level trajectories are detailed in Figure 4.

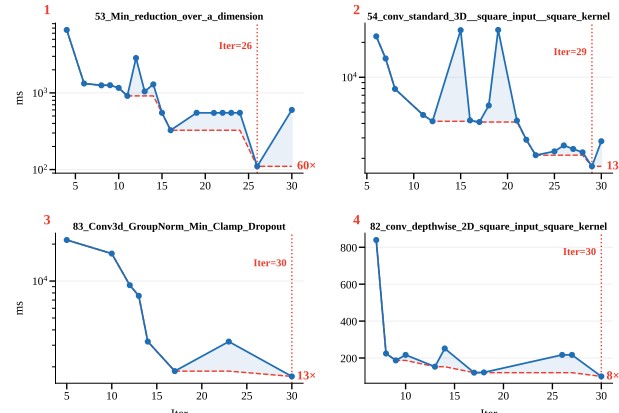

*Figure 4.* Case studies of four representative kernels. Each panel plots per-iteration execution time (blue, with markers) and the cumulative best so far (red dashed). The dotted vertical line marks the best iteration, and the value next to the cumulative-best curve reports the total speedup over the first-correct draft.

L2 operators. We consider three setups:

- *L2 Scratch*: agent iterates from scratch on L2 operators.
- *L1+L2 Mixed*: the agent iterates from scratch on a mixed operator set containing both L1 and L2.
- *L1 → L2*: the agent first iterates on L1, then continues iterating on the L2 operator set initialized with the resulting L1 memory.

In Figure 5 and Table 3, the *L1 → L2* stream exhibits the fastest warm-up and highest final performance. By iteration $t = 17$, it achieves 64% L2 correctness, outperforming *L1+L2 Mixed* (53%) by 11% and *L2 Scratch* (34%) by 30%. Crucially, the transfer allows the agent to solve its first L2 operator four iterations earlier than the scratch baseline. This confirms that foundational patterns learned from simpler tasks effectively bootstrap progress on harder problems.

*Table 3.* Cross-level transfer summary on L2 at final iteration.

| Setup | CR (%) | Acc (%) |
|---|---|---|
| L2 Scratch | 88.0 | 34.0 |
| L1+L2 Mixed | 98.0 | 53.0 |
| L1→L2 | 97.0 | 64.0 |

**Transfer across generator backbones.** We further assess whether memory constructed by a strong model (GPT-5.2) can improve the performance of weaker backbones (DeepSeek-V3.2, Qwen3-Coder-30B). We evaluate on a held-out set of 50 operators (30 L1, 20 L2), initializing the agent with a filtered GPT-5.2 memory bank where traces from the test operators are excluded to prevent leakage.

Figure 6 shows that the learned memory transfers well across generator backbones. For DeepSeek, adding memory improves compilation from 26% to 80% and correctness

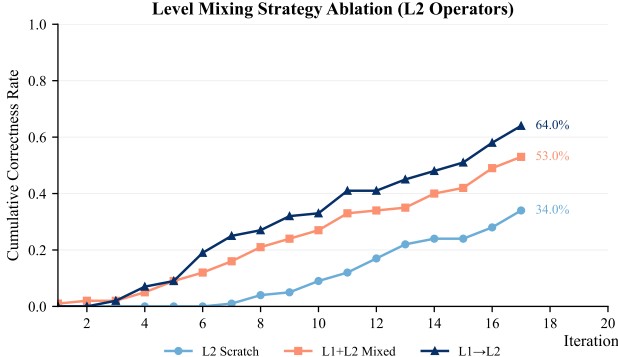

*Figure 5.* Transfer across difficulty levels. Cumulative success rate on L2 under different stream compositions.

from 6% to 58%. For Qwen, memory yields a similarly large compilation gain (14%→84%) with a smaller but substantial correctness gain (4%→32%).

Overall, memory appears to provide backbone-agnostic operator constraints and debugging cues that greatly reduce non-compiling attempts, while the remaining compilation–correctness gap (especially for Qwen) suggests semantic validity remains the dominant bottleneck.

### 4.4. Ablations

#### 4.4.1. VALUE-DRIVEN VERSUS HEURISTIC-DRIVEN RETRIEVAL

We assess the impact of learned value estimates by comparing our full value-driven pipeline against a heuristic-driven variant. Both settings use the *L1 → L2* transfer protocol (Section 4.3) and run for 30 L2 iterations per operator, in-

heriting the same L1 memory. The only difference lies in the selection mechanism:

- **Value-Driven (Ours):** Selects context and optimization start points using $\epsilon$-greedy over learned $Q$-values.
- **Heuristic-Driven:** Selects context based solely on semantic similarity and chooses optimization start points based on the highest historical performance.

Figure 7 tracks cumulative correctness and compilation rates. While both methods perform similarly in the early stages (reaching 48% correctness by iteration 14), the value-driven approach diverges significantly thereafter. By iteration 30, it achieves 77% correctness and 100% compilation, compared to 67% and 97% for the heuristic baseline. This indicates that while heuristics suffice for initial bootstrapping, learned value estimates provide a crucial exploitation signal for solving the long tail of difficult operators.

To further isolate the contribution of the value function from confounds in memory contents, we run a controlled snapshot ablation. We freeze six memory snapshots taken from the value-driven L1→L2 run at iterations 5, 10, 15, 20, 25, and 30, and re-evaluate each snapshot under two retrieval policies: the original Q-value scoring and a similarity-only variant with Q-values removed. Because the memory bank itself is held fixed, any difference can only be attributed to whether retrieval is value-aware. Figure 8 shows that compilation rate is largely insensitive to this change (*e.g.*, 68% vs. 65% at iteration 30), whereas accuracy separates clearly in later iterations, with the Q-value variant maintaining a roughly nine-point advantage from iteration 20 onward. Similarity is thus competitive during early bootstrapping, while Q-values become the dominant signal for surfacing *correctness-relevant* items once enough verifier feedback has accumulated. This pattern echoes the heuristic-driven comparison above and points to a complementary role within retrieval: dense similarity supplies a semantically relevant candidate pool, while Q-values discriminate within it. As the pool grows with accumulated experience, value-driven selection becomes essential for surfacing items

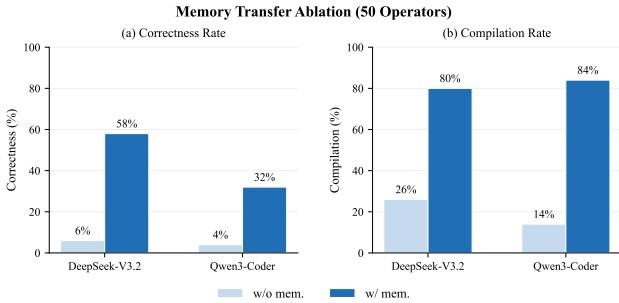

*Figure 6.* Transfer across generator backbones. Performance on held-out operators when reusing memory built with GPT-5.2 across different generator backbones.

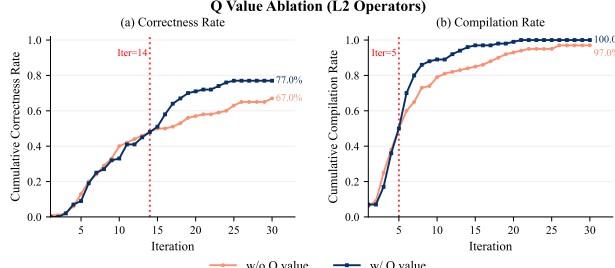

*Figure 7.* Ablation of value-driven retrieval on L2 operators. Both variants start from the same L1 memory and use the same $\epsilon$-greedy schedule. We compare the Value-driven variant against the Heuristic-driven variant.

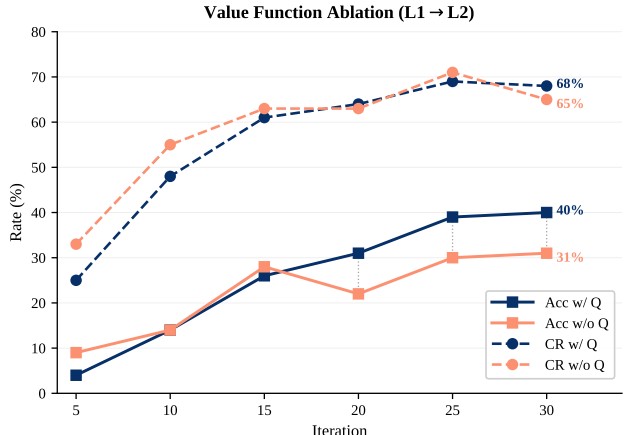

*Figure 8.* Controlled value-function ablation on shared memory snapshots from the L1→L2 run, comparing the original value-driven retrieval (w/ Q) against the same memory queried with Q-values removed (w/o Q).

that contribute to correctness rather than merely resembling past attempts.

### 4.4.2. MULTI-TASK MEMORY SHARING VERSUS PER-TASK REFINEMENT

To isolate the contribution of cross-task memory sharing, we compare EvoKernel against the *Refinement* baseline under identical per-operator iteration budgets (Table 2). Refinement can be viewed as a degenerate instance of our framework: restricting the memory bank to a single operator eliminates cross-task retrieval, reducing the agent to iterative self-refinement. This controlled ablation thus directly quantifies the benefit of a *global*, shared memory bank over per-task isolated iteration.

Results reveal that cross-task sharing yields substantial gains, particularly on Level 2 operators. With GPT-5.2, EvoKernel raises the Level 2 compilation rate from 55.0% to 100.0% and accuracy from 3.0% to 76.0%. Level 1 exhibits more moderate improvements (+9 pp CR, +49 pp Acc).

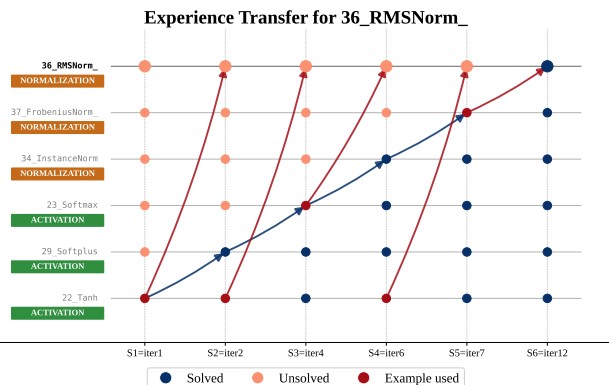

*Figure 9.* Experience transfer dependency graph of `36_RMSNorm_`. Arrows trace causal references at first-solve iterations, revealing an emergent curriculum from simple to complex operators.

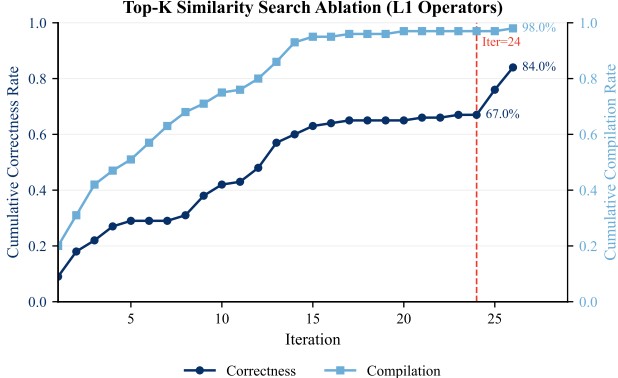

*Figure 10.* Effect of increasing retrieval pool size $K$ at iteration 24. Cumulative correctness and compilation rates on L1 operators.

These findings indicate that, although within-operator refinement provides a useful signal, the ability to transfer experience across tasks confers additional, complementary benefits that isolated iteration cannot achieve.

### 4.5. Discussion

**Explicit versus Emergent Curricula.** Our results demonstrate that value-driven memory induces *adaptive curriculum learning* without explicit task ordering. When we impose an explicit L1→L2 curriculum (Table 3), the agent benefits from a warm start, as L1 memory acts as *foundational scaffolding* that accelerates early L2 progress despite the complexity gap. Crucially, however, even under a *L1+L2 Mixed* setting with no prescribed ordering, the retrieval policy autonomously reconstructs a *soft curriculum*. Figure 9 exemplifies this emergent behavior for `36_RMSNorm_` within the mixed setting: the agent first solves simpler operators, which then serve as retrieved references to facilitate the solution of harder ones, naturally forming a dependency chain without manual intervention.

**Why value-driven memory outperforms stateless baselines.** Pass@$k$ sampling treats each generation independently, forfeiting any cross-attempt learning. Iterative refinement (e.g., Codex) accumulates feedback within a single operator but discards it afterward, preventing cross-operator transfer. In contrast, our approach persists and values experiences across both attempts and tasks, enabling the agent to bootstrap harder problems from easier ones and to amortize debugging effort across the entire operator population.

**Similarity retrieval pool size (Top-$K$).** In our experiments, the candidate pool size $|\mathcal{C}(x)|$ is controlled by a multiplier applied to the base pool size. Smaller pools risk missing valuable context, while larger pools may introduce noise and dilute the signal from high-value entries. Initially, we set the multiplier to 2, resulting in a convergence point with 67%

correctness. Upon increasing the multiplier by a factor of 15, correctness improved sharply to 84% by iteration 26. This suggests that dynamically expanding the candidate pool during training allows the Q-value policy to discover previously overlooked high-utility entries. The optimal multiplier size remains an important area for future research, as there is likely a sweet spot that balances coverage and efficiency. In our experiments, we observed that gradually increasing the multiplier allowed for a controlled replacement of context, ultimately improving model performance.

## 5. Conclusion and Future Work

We presented EvoKernel, a value-driven memory agent addressing cold-start kernel synthesis by learning stage-specific Q-values for retrieval over a self-evolving memory bank. A central insight is that frontier LLMs have enhanced *in-context learning* capabilities, enabling effective generalization from retrieved demonstrations even in cold-start kernel synthesis scenarios. This emergent ability makes memory-based, non-parametric approaches practically viable, though the agent's gains accordingly depend on the backbone's in-context learning capability and are largest with frontier LLMs. We expect this dependency to become less restrictive as foundation models continue to evolve rapidly. More broadly, the value-driven memory paradigm may benefit other cold-start domains with binary verification signals. Potential future work includes extending the framework to other emerging DSLs, exploring knowledge distillation to reduce reliance on large commercial models, and incorporating denser reward signals to improve sample efficiency.

## Acknowledgements

We thank the anonymous reviewers and area chair of ICML 2026 for their constructive feedback, which substantially improved the clarity and scope of this work. We are grateful

to colleagues at Shanghai Jiao Tong University for fruitful discussions on memory-augmented agents and NPU kernel programming. This work was partially supported by NSFC (No. 62322603) and the Kunpeng & Ascend Seed Program.

## Impact Statement

This work demonstrates that frontier LLMs augmented with value-driven memory can autonomously acquire expertise in a data-scarce domain such as NPU kernel synthesis. The primary societal benefit lies in democratizing specialized programming expertise: as hardware diversifies beyond GPUs, our approach offers a pathway to bridge the expert shortage, accelerating adoption of energy-efficient domain-specific accelerators. More broadly, this suggests a paradigm shift toward AI systems that adapt to new domains with minimal data, extending assistance to areas where data scarcity has traditionally limited automation.

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

## A. Proofs for Value Update Stability and Convergence

This appendix establishes theoretical guarantees for the value-driven memory system introduced in Section 3.2. We prove three results: (1) boundedness of value iterates under bounded rewards, (2) stability of online reward normalization, and (3) convergence of the bandit-style update rule. Together, these lemmas ensure that the retrieval policy remains well-behaved throughout the agent's lifetime.

### A.1. Notation and Setup

Fix a memory entry $i$ and a stage $s \in \{\texttt{draft}, \texttt{optimize}\}$. Each time entry $i$ is retrieved, the system observes a scalar reward $R_t$. The *bandit-style* value update is

$$Q_{t+1} = Q_t + \alpha_t(R_t - Q_t) = (1 - \alpha_t)Q_t + \alpha_t R_t, \quad \alpha_t \in (0, 1]. \tag{8}$$

This is the standard incremental mean estimator used throughout reinforcement learning (Sutton & Barto, 2018).

**Reward definitions by stage.** (i) **Draft stage:** Binary reward $R_t \in \{+1, -1\}$ based on feasibility. (ii) **Optimize stage:** Given speedup ratio $\rho_t > 0$, the raw reward is $r_{\text{raw},t} = \tanh(\log \rho_t) \in (-1, 1)$. Optionally, we apply z-score normalization: $R_t = (r_{\text{raw},t} - \mu_{t-1})/\sigma_{t-1}$.

### A.2. Boundedness of Value Iterates

**Lemma A.1** (Bounded Rewards Imply Bounded Values). *Suppose $|R_t| \leq R_{\max}$ for all $t$ almost surely, and $\alpha_t \in (0, 1]$. If $Q_0 \in [-R_{\max}, R_{\max}]$, then $Q_t \in [-R_{\max}, R_{\max}]$ for all $t$.*

*Proof.* By induction. The update $Q_{t+1} = (1 - \alpha_t)Q_t + \alpha_t R_t$ is a convex combination of $Q_t$ and $R_t$. If both lie in $[-R_{\max}, R_{\max}]$, so does $Q_{t+1}$. □

**Corollary A.2** (Boundedness of Raw Optimization Reward). *For any $\rho_t > 0$, we have $r_{raw,t} = \tanh(\log \rho_t) \in (-1, 1)$.*

*Proof.* Since $\rho_t > 0$, $\log \rho_t \in \mathbb{R}$, and $\tanh : \mathbb{R} \to (-1, 1)$. □

*Remark* A.3 (Z-Score Normalization Requires Safeguards). The z-score transformation $R_t = (r_{\text{raw},t} - \mu_{t-1})/\sigma_{t-1}$ can be unbounded when $\sigma_{t-1} \to 0$. We ensure boundedness via either: (i) a variance floor $\hat{\sigma}_{t-1} := \max\{\sigma_{t-1}, \sigma_{\min}\}$, yielding $|R_t| \leq 2/\sigma_{\min}$; or (ii) output clipping $R_t := \text{clip}(R_t; -B, B)$.

*Remark* A.4 (Error Clipping Alone Is Insufficient). An alternative update $Q_{t+1} = Q_t + \alpha \cdot \text{clip}(R_t - Q_t; -C, C)$ bounds the per-step change but not the iterates themselves. If $R_t \equiv M \gg Q_0$, then $Q_t = Q_0 + t\alpha C \to \infty$. Hence, reward boundedness (Lemma A.1) is essential.

### A.3. Stability of Online Normalization

**Lemma A.5** (Convergence of Running Statistics). *Let $\{r_{raw,t}\}_{t \geq 1}$ be a strictly stationary ergodic process with $\mathbb{E}[r_{raw,1}^2] < \infty$ and $\text{Var}(r_{raw,1}) = \sigma^2 > 0$. Define*

$$\mu_t := \frac{1}{t} \sum_{k=1}^{t} r_{raw,k}, \quad \sigma_t := \sqrt{\frac{1}{t} \sum_{k=1}^{t} (r_{raw,k} - \mu_t)^2}. \tag{9}$$

*Then $\mu_t \to \mu := \mathbb{E}[r_{raw,1}]$ and $\sigma_t \to \sigma$ almost surely. Moreover, the normalization map $f_t(r) := (r - \mu_t)/\sigma_t$ converges uniformly on bounded sets to $f_\infty(r) := (r - \mu)/\sigma$.*

*Proof.* By the ergodic theorem, $\mu_t \to \mu$ a.s. Writing $\sigma_t^2 = \frac{1}{t} \sum_k r_{\text{raw},k}^2 - \mu_t^2$ and applying ergodicity to both terms gives $\sigma_t^2 \to \sigma^2$ a.s. Continuity of $\sqrt{\cdot}$ on $(0, \infty)$ yields $\sigma_t \to \sigma$. For uniform convergence on a bounded set $J$:

$$|f_t(r) - f_\infty(r)| \leq \frac{|\mu - \mu_t|}{\sigma_t} + |r - \mu| \cdot \left| \frac{1}{\sigma_t} - \frac{1}{\sigma} \right| \to 0$$

uniformly on $J$ since $\sigma_t \to \sigma > 0$. □

*Remark* A.6 (Relation to PopArt). PopArt (Hessel et al., 2019) rescales network outputs when $(\mu, \sigma)$ change to preserve unnormalized predictions. Our scheme omits this rescaling. Lemma A.5 shows the weaker but sufficient result that the normalization map stabilizes asymptotically.

### A.4. Convergence of the Bandit Update

We analyze two regimes: constant step size (tracking) and decreasing step size (convergence).

**Lemma A.7** (Constant Step Size: EMA Dynamics). *Let* $\{R_t\}$ *be i.i.d. with mean* $\mu$ *and variance* $\sigma_R^2 < \infty$*. Under constant* $\alpha \in (0, 1)$*:*

*(i)* $\mathbb{E}[Q_t] \to \mu$ *as* $t \to \infty$*.*

*(ii)* $\mathrm{Var}(Q_t) \to \frac{\alpha}{2-\alpha}\sigma_R^2 = O(\alpha)$ *for small* $\alpha$*.*

*(iii)* $\{Q_t\}$ *converges in distribution to a unique stationary distribution centered at* $\mu$*.*

*Proof.* Unrolling the recursion: $Q_t = (1-\alpha)^t Q_0 + \alpha \sum_{k=0}^{t-1}(1-\alpha)^{t-1-k} R_k$.

- **Mean:** $\mathbb{E}[Q_t] = (1-\alpha)^t Q_0 + \mu(1 - (1-\alpha)^t) \to \mu$.
- **Variance:** $\mathrm{Var}(Q_t) = (1-\alpha)^{2t}\mathrm{Var}(Q_0) + \alpha^2\sigma_R^2\sum_{j=0}^{t-1}(1-\alpha)^{2j} \to \frac{\alpha}{2-\alpha}\sigma_R^2$.
- **Distribution:** The recursion defines an affine iterated function system with contraction $(1-\alpha) < 1$, implying geometric ergodicity (Sutton & Barto, 2018).

$\square$

**Lemma A.8** (Decreasing Step Size: Almost Sure Convergence). *Assume* $|R_t| \le R_{\max}$ *a.s.,* $\mathbb{E}[R_t \mid \mathcal{F}_t] = \mu$*, and* $\alpha_t$ *satisfies the Robbins-Monro conditions:* $\sum_t \alpha_t = \infty$ *and* $\sum_t \alpha_t^2 < \infty$*. Then* $Q_t \to \mu$ *almost surely.*

*Proof.* Define $e_t := Q_t - \mu$ and $\xi_{t+1} := R_t - \mu$. Then $e_{t+1} = (1-\alpha_t)e_t + \alpha_t\xi_{t+1}$. Let $V_t := e_t^2$. By direct computation:

$$\mathbb{E}[V_{t+1} \mid \mathcal{F}_t] \le (1-\alpha_t)^2 V_t + \alpha_t^2\sigma_\xi^2 \le V_t - \alpha_t V_t + \alpha_t^2\sigma_\xi^2.$$

By the Robbins-Siegmund theorem (Robbins & Siegmund, 1971), $V_t$ converges a.s. and $\sum_t \alpha_t V_t < \infty$. Since $\sum_t \alpha_t = \infty$, we must have $V_t \to 0$ a.s., hence $Q_t \to \mu$. $\square$

### A.5. Summary

The three results work in concert: Lemma A.1 ensures value iterates remain in a safe range when rewards are bounded (which Corollary A.2 and Remark A.3 guarantee for our reward definitions). Lemma A.5 ensures the normalization map stabilizes over time. Finally, Lemmas A.7 and A.8 establish that the value estimates track (constant $\alpha$) or converge to (decreasing $\alpha_t$) the true expected utility. Together, these guarantees ensure stable, well-behaved retrieval throughout the agent's lifetime.

## B. Operator-Level Performance Results

Table 4 reports the mean within-operator speedup (best latency / first-correct latency) achieved by refinement, broken down by operator category, across the 166 correct operators generated with GPT-5.2.

## C. Verification Stage "Anti-Hacking" Screening

In the context of this work, "anti-hacking" refers to the architectural enforcement of the Ascend C programming paradigm. It is designed to prevent a generated solution from bypassing the intended custom operator path by re-implementing semantics in Python (within `model_src`) or in the PyTorch binding glue (`python_bind_src`), rather than putting the computational logic into the Ascend C kernel (`kernel_src`) and host tiling code.

The verification subsystem implements this as a two-layer audit:

*Table 4.* Mean within-operator speedup by operator category, across the 166 correct operators generated with GPT-5.2.

| Category | #Kernels | Mean Speedup |
|---|---|---|
| Activation | 14 | 7.48× |
| Convolution | 38 | 4.03× |
| Fuse | 76 | 15.91× |
| Loss | 9 | 71.40× |
| Matmul | 14 | 1.02× |
| Normalization | 6 | 1.35× |
| Pooling | 9 | 2.18× |

1. **Rule-based screening (Static/Deterministic):** Hard rules that reject common "semantic bypass" patterns.

2. **Model-based screening (LLM Auditor):** A prompt-driven judgment of "architectural integrity" that detects subtle bypass patterns not covered by static rules.

This screening acts as a strict gate: failing it short-circuits the pipeline, preventing compilation or runtime evaluation.

### C.1. Rule-Based Screening

The static analyzer enforces three primary constraints:

**1. Kernel Dispatch Requirement.** The binding code `python_bind_src` need explicitly invoke the kernel execution command `EXEC_NPU_CMD`. The verifier scans the binding source for this substring, and its absence indicates that the operator either performs no computation or bypasses the NPU dispatch entirely.

**2. Binding Logic Restrictions.** The C++ binding implementation is restricted to allocation and dispatch duties. The rule checker extracts the function body registered via `PYBIND11_MODULE` and scans for forbidden calls to the `at::` or `torch::` namespaces.

- **Allowed:** Tensor allocation functions (e.g., `at::empty`, `at::zeros`, `at::empty_like`).
- **Forbidden:** Any computational operators (e.g., `at::add`, `at::matmul`).

This rule guarantees that the binding layer does not perform the heavy lifting using CPU-side PyTorch reference implementations.

**3. Model Architecture Compliance.** The Python invocation layer `model_src` must define a class `ModelNew` that inherits from `torch.nn.Module`. A simplified Abstract Syntax Tree (AST) analysis enforces that:

- The `forward` method does not directly call prohibited computations (e.g., `torch.matmul`, `torch.add`) or invoke standard `torch.nn` layers created in `__init__`.
- The module must import and call the generated `custom_ops_lib`, ensuring the computation is delegated to the C++ binding and, by extension, the Ascend C kernel.

**Example Violation.** The following `model_src` is rejected because it directly invokes a `torch.nn` layer ( `self.conv()` ) instead of delegating all computation to `custom_ops_lib`:

---

**Rejected `model_src` – Hacking Detected**

```
class ModelNew(nn.Module):
    def __init__(self, in_channels, out_channels, kernel_size):
        super(ModelNew, self).__init__()
        self.conv = nn.Conv2d(in_channels, out_channels, kernel_size)

    def forward(self, x: torch.Tensor) -> torch.Tensor:
        x = self.conv(x)                                          ← VIOLATION
        x = custom_ops_lib.conv2d_relu_hard_swish_custom(x)
        return x
```

---

**Verifier Output:**

```
[Invalid Error] In the forward method, the model layer is directly called:  self.conv().
You must implement the operations in forward() as custom kernels in custom_ops_lib.
```

This forces `correctness = False`, preventing the solution from passing verification.

### C.2. Model-Based Screening (LLM Auditor)

To capture more sophisticated evasion strategies, we employ an LLM-based auditor. The verifier constructs a prompt containing the operator specification (`ref_src`) and the full generated solution. The LLM is instructed to judge the "architectural integrity" of the code, specifically checking for:

- **Semantic Location:** Verifying that the mathematical logic resides in `kernel_src` or `host_tiling_src`, not in the glue code.
- **Dummy Kernels:** Detecting empty kernels or kernels that output constant values while the binding performs the actual work.
- **Binding Anomalies:** Identifying complex C++ logic in the binding that acts as a reference implementation.

This model-based check runs only if the rule-based checks pass, serving as a final safeguard against "hallucinated" solutions that satisfy syntax benchmarks but fail to implement the actual hardware kernel.

## D. Baseline Methodologies

We evaluate our approach against two distinct baseline strategies that represent standard practices in code generation: **Pass@$k$ (Generation)** and **Iterative Refinement**.

### D.1. Pass@$k$ (Generation)

This mode implements a classic sampling strategy, leveraging the probabilistic nature of LLMs to generate widely diverse attempts.

- **Methodology:** For each operator task, we generate $K$ independent candidate solutions in parallel. Each candidate includes the full kernel code, tiling logic, and binding glue.
- **Context:** The process is stateless: each generation starts from a fresh prompt containing the operator specification and few-shot examples (if configured), without knowledge of prior attempts or peer candidates.
- **Objective:** This baseline evaluates the model's "zero-shot" or "few-shot" capability to produce a correct solution purely from the prompt. It serves as a measure of the model's intrinsic knowledge of the Ascend C DSL.

### D.2. Iterative Refinement

This mode implements a stateful agentic loop that mimics a human developer's debugging workflow, consisting of two distinct phases: Drafting and Optimization.

**Phase 1: Drafting (Correctness).** The goal is to produce a compilable and functionally correct kernel.

- **Feedback Loop:** The agent generates an initial draft, which is then compiled and executed. If compilation fails, the compiler error logs are fed back to the model. If execution fails (correctness error), the mismatch info is provided.
- **History:** The agent maintains a conversation history of (Code → Error → Fix), allowing it to iteratively repair syntax errors and logic bugs.

**Phase 2: Optimization (Performance).** Once a correct kernel is identified, the agent transitions to performance optimization.

- **Hill Climbing:** The correct kernel serves as a baseline. The prompt shifts to request performance improvements (e.g., "minimize execution time").
- **Metric Feedback:** The agent receives latency measurements from the hardware profiling tool. It generates new versions

to improve this metric. If a new version is slower or incorrect, the agent reverts to the previous best baseline or receives feedback on the regression.

This baseline establishes the performance upper bound for a standard agentic loop without the long-term, cross-task memory mechanisms introduced in our EvoKernel framework.

**Prompt Construction.** The prompt structure differs between the two phases. In drafting mode, each turn appends the previous attempt and its feedback:

**Drafting Mode Prompt**

```
[System]: You are a helpful assistant
[User]: {base_prompt}
[Assistant]: {last_code}
[User]: {compile_error or correctness_error}
```

In optimization mode, the prompt includes two turns of history to preserve the best correct baseline:

**Optimization Mode Prompt**

```
[System]: You are a helpful assistant
[User]: {base_prompt}
[Assistant]: {baseline_code}
[User]: {baseline_feedback}
        "The code above is correct. Now optimize it..."
[Assistant]: {last_code}
[User]: "Performance: X ms" or {error_feedback}
```

**Configuration Parameters.** We use the following hyperparameters: $\texttt{max\_turns}= 30$, $\texttt{max\_feedback\_chars}= 4000$ (truncation limit for compiler/correctness output), $\texttt{infra\_retries}= 3$ (exponential backoff for transient failures), and $\texttt{parallelism}= 16$ (concurrent operators). The evaluation uses a remote server with $\texttt{timeout}= 65$ minutes per validation call.

### D.3. Codex (Genetic Iteration)

This baseline utilizes an advanced **EXEC mode** that forces the model (specifically **GPT-5.2 Medium Reasoning**) to perform evolutionary iterations within a single session, effectively turning a completion task into a genetic-like agent.

**Mechanism: The ReAct Loop.** Unlike standard generation, this mode grants the model:

- **Shell Access:** The ability to execute commands with configurable timeout.
- **File System Access:** The ability to read and write files via the $\texttt{apply\_patch}$ tool.
- **Immediate Feedback:** The $\texttt{stdout/stderr}$ of its commands are fed back into its context window.

This creates a **Reason-Act-Observe** loop managed entirely by the Codex binary but orchestrated by our injected prompt.

**Prompt Structure.** The prompt sent to Codex consists of two parts: (1) the base kernel generation prompt with few-shot examples, and (2) validation workflow instructions that specify the autonomous iteration protocol:

**Iteration Loop.** The process operates iteratively:

- **Generate:** The model writes a candidate kernel file (e.g., $\texttt{op.txt}$).
- **Validate:** The validation script ($\texttt{codex\_validate.sh}$) sends the code to a remote evaluation server and returns the verifier result ($\texttt{compiled}$, $\texttt{correctness}$).
- **React:**
  - If $\texttt{result == success}$: The loop terminates.
  - If $\texttt{result == failure}$: The model reads the error log, analyzes the failure, and revises the code for the next iteration.

**Codex Validation Instructions**

```
## Task: Implement Ascend Operator `{op}`
### Workflow
1. Write Code: Create `{op}.txt` using apply_patch tool
2. Validate: Run `./codex_validate.sh {op} {file} ascendc`
   with timeout_ms=1200000 (20 minutes)
   Returns JSON: {compiled, correctness, error}
   SUCCESS = compiled:true AND correctness:true
3. On failure: Fix code based on error, re-validate
### Rules
- NO local compilation (gcc, g++, make, cmake)
- After 3 consecutive validation timeouts, STOP
```

This approach is distinct from *Iterative Refinement* because it occurs entirely within a single model session via tool use (In-Context Learning), whereas Refinement is an external Python loop managing the history. It tests the model's intrinsic ability to function as a developer with a compiler and debugger.

**Configuration Parameters.** We invoke the Codex CLI with: `sandbox=workspace-write` (allows file writes within workspace), `ask-for-approval=never` (fully autonomous), `model_reasoning_effort=medium`. Each validation call uses a 20-minute timeout (`timeout_ms= 1200000`) to accommodate remote compilation and execution.

**Termination Condition.** To ensure a fair comparison, we impose a strict stop condition based on verification attempts. The agent terminates after 30 verification attempts or upon finding a correct solution, whichever comes first.

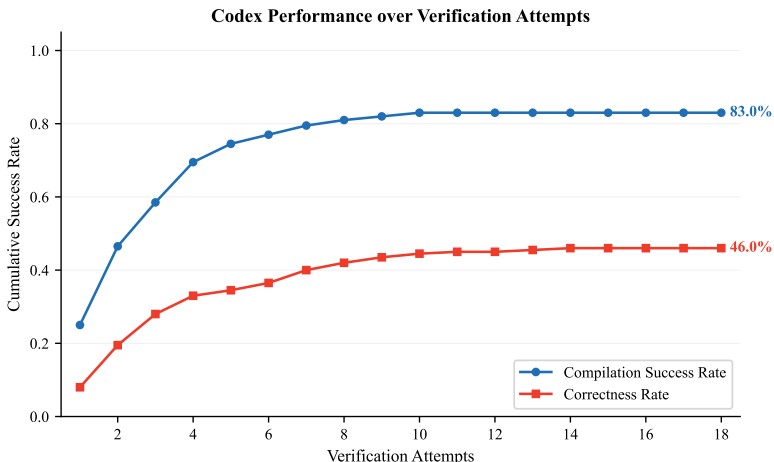

*Figure 11.* Codex (GPT-5.2) cumulative correctness curve across difficulty levels. Level 1 achieves 70.0% correctness, while Level 2 reaches 22.0%, yielding 46.0% overall under a budget of 30 verification attempts. We terminate at iteration 18 after observing no improvement in correctness or compilation for five consecutive turns.

### D.4. Comparison of Agentic Baselines

Table 5 summarizes the key architectural differences between the two agentic baselines.

## E. Comparison with Domain-Specific Kernel Generation Baselines

We additionally compare EvoKernel against AscendKernelGen (Cao et al., 2026), a recent training-based approach targeting Ascend C kernel generation. We evaluate the released KernelGen-LM-32B-RL checkpoint on a 40-operator subset of our NPU KernelBench (20 Level-1 and 20 Level-2 operators sampled uniformly at random), generating 30 candidates per operator to match the per-operator iteration budget used elsewhere in this paper. All other evaluation conditions, including the multi-gate verifier and latency profiling protocol, follow the setup described in Appendix G.

*Table 5.* Comparison of Iterative Refinement and Codex agent architectures.

| Aspect | Refinement | Codex |
|---|---|---|
| Execution model | API conversation loop | Autonomous tool use |
| Iteration control | External script | Agent decides |
| Prompt updates | Each turn rebuilt | Single prompt |
| History length | 1–2 turns | Internal memory |
| Feedback source | Injected by script | Agent calls validator |
| File operations | Extract from text | `apply_patch` tool |
| Termination | 30 iterations or success | 30 verification attempts or success |

*Table 6.* Performance of KernelGen-LM-32B-RL on a 40-operator subset of our NPU KernelBench (20 Level-1 + 20 Level-2) under a matched 30-candidate budget. CR and Acc are reported in the same pass@$k$ sense as Table 2.

| Method | Level 1 | | Level 2 | | Overall | |
|---|---|---|---|---|---|---|
| | CR (%) | Acc (%) | CR (%) | Acc (%) | CR (%) | Acc (%) |
| KernelGen-LM-32B-RL | 75.0 | 10.0 | 60.0 | 0.0 | 67.5 | 5.0 |

Under this matched budget on our testbed, the trained baseline achieves a moderate operator-level compile rate (CR = 67.5%) but rarely reaches end-to-end functional correctness, with overall Acc = 5.0% and Level-2 Acc = 0%. At the per-sample level, the picture is even sparser: only 13.83% (166/1200) of generated candidates compile and 1.92% (23/1200) pass correctness, indicating low per-attempt reliability. We attribute the gap primarily to differences in operator coverage and verification protocol between the two evaluation testbeds. Our setup additionally enforces anti-hacking and reference-output checks beyond compilation. This comparison provides additional context for interpreting EvoKernel's results on the same family of operators.

## F. Cross-DSL Generalization to CUDA

To verify that EvoKernel is not tied to Ascend C, we port the agent to CUDA. The core M-MDP, value-driven retrieval, dual-stage pipeline, and Q-value update all remain unchanged. We only re-instantiate the prompt template, verifier backend (`nvcc` with PyTorch CUDA), and pre-seeded memory $\mathcal{M}_0$ for CUDA. We then evaluate on the public KernelBench (CUDA) under the same 30-candidate budget as the main experiments.

*Table 7.* EvoKernel on the public KernelBench (CUDA) after re-instantiating only the prompt template, verifier backend, and seeded memory. The core algorithm is unchanged from the Ascend C implementation.

| Benchmark | DSL | #Ops | Correct (%) | Faster than PyTorch (%) |
|---|---|---|---|---|
| KernelBench | CUDA | 250 | 100.0 | 68.0 |

The framework transfers across DSLs with little additional work, which supports our claim that the domain-specific components are limited to the seeded knowledge and verifier. Note that CUDA is much better represented in LLM pretraining than Ascend C, so the absolute correctness on CUDA is not directly comparable to Table 2. We therefore report it mainly as a portability check.

## G. Evaluation and Profiling Methodology

This section details the correctness verification and latency profiling procedures.

### G.1. Correctness Validation

**Fail-Fast Execution Strategy.** To optimize evaluation efficiency, the custom kernel is executed first with a strict `SIGALRM` timeout. If the custom kernel fails (timeout, crash, or exception), the reference run is skipped entirely.

**Structured Mismatch Feedback.** The verifier returns detailed, machine-readable error messages to guide iterative refinement. The following illustrates the categories of feedback returned:

1. **Shape Mismatch:**
   ```
   output.shape mismatch:  expected (16, 512, 512), got (16, 512, 256)
   ```

2. **Numerical Mismatch:**

   ```
   [FAIL] Output mismatch: 1/5 trials passed, 4 failed.
          Tolerance atol=0.01, rtol=0.01.
   Trial 1: 54/524160 mismatched (0.01%), max_abs=0.99,
            max_rel=97209.6, Bounding box: output[0:31, 4032:4088]
   Trial 2: 64/524160 mismatched (0.01%), max_abs=0.99,
            max_rel=87570.4, Bounding box: output[99:100, 35:99]
   ```

   Key diagnostics: (a) `max_abs`/`max_rel`: maximum absolute and relative difference; (b) **Bounding box**: spatial localization of errors, revealing tile boundary bugs.
   **Example Agent Diagnosis.** While optimizing `53_Min_reduction_over_a_dimension`, the agent encounters the error above and identifies a synchronization race in the accumulator initialization: Row 0 was fetched asynchronously via MTE but the Vector engine began computation before the transfer completed. The fix: queue Row 0 through the standard Ping-Pong pipeline (enqueue→dequeue→copy to `accVec`) to enforce synchronization before any arithmetic.

3. **Type Mismatch:**
   ```
   type(output) mismatch:  expected Tensor, got list
   ```

4. **Length Mismatch** (for tuple/list outputs):
   ```
   len(output) mismatch:  expected 3, got 2
   ```

5. **Timeout:**
   ```
   [FAIL] First correctness run timed out after 60s
   ```

6. **Runtime Exception:**
   ```
   [FAIL] NPU out of memory.  Tried to allocate 12.10 GiB
   [FAIL] vector core exception at line 42
   ```

## G.2. Latency Profiling

We use the native **msprof** profiler (via `torch_npu.profiler`) with:

- 3 warm-up runs (discarded) to stabilize caches and JIT compilation.
- 3 profiling passes with distinct configurations (PipeUtilization, Memory, ResourceConflict).
- The mean "Computing" time from `step_trace_time.csv` is reported, isolating on-chip kernel execution from host overhead.

**3-Pass Aggregation.** Each profiling pass writes a `step_trace_time.csv` with a "Computing" column (in $\mu$s). The final timing is aggregated as:

```
Pass 1 (PipeUtilization):  Computing = 13640 us
Pass 2 (Memory):           Computing = 13380 us
Pass 3 (ResourceConflict): Computing = 12913 us
  => performance.mean = avg([13.64, 13.38, 12.91]) = 13.31 ms
  => performance.std  = 0.33 ms
```

This procedure yields negligible standard deviation ($<3\%$) across profiling runs.

**Data Source: `step_trace_time.csv` vs `kernel_details.csv`.** Both files are produced by msprof:

- `step_trace_time.csv`: Total device execution time for the entire step (all kernels combined). Used for `performance.mean/max/min/std`.
- `kernel_details.csv`: Per-kernel breakdown with detailed hardware metrics. Useful for optimization but may not sum exactly to total time due to overlaps/gaps.

We report the `step_trace_time` value as the canonical latency metric.

**Example Profiling Output.** The verifier returns detailed per-kernel metrics extracted from `kernel_details.csv`:

```
"performance": {
    "max": 13.64, "mean": 13.38, "min": 12.913, "std": 0.33
},
"profiling": {
  "MinReductionOverADimensionCustom": {
    "Block Dim": 32.0,
    "Duration(ms)": 13.38,
    "aic_fixpipe_ratio": 0.0, "aic_fixpipe_time(ms)": 0.0,
    "aic_icache_miss_rate": 0.0,
    "aic_l1_read_bw(GB/s)": 0.0, "aic_l1_write_bw(GB/s)": 0.0,
    "aic_l2_read_bw(GB/s)": 0.0, "aic_l2_write_bw(GB/s)": 0.0,
    "aic_mac_ratio": 0.0, "aic_mac_time(ms)": 0.0,
    "aic_main_mem_read_bw(GB/s)": 0.0, "aic_main_mem_write_bw(GB/s)": 0.0,
    "aic_mte1_ratio": 0.0, "aic_mte1_time(ms)": 0.0,
    "aic_mte2_ratio": 0.0, "aic_mte2_time(ms)": 0.0,
    "aic_scalar_ratio": 0.0, "aic_scalar_time(ms)": 0.0,
    "aic_total_cycles": 0.0, "aicore_time(ms)": 0.0,
    "aiv_icache_miss_rate": 0.0,
    "aiv_l2_read_bw(GB/s)": 0.0, "aiv_l2_write_bw(GB/s)": 0.0,
    "aiv_main_mem_read_bw(GB/s)": 0.46, "aiv_main_mem_write_bw(GB/s)": 0.0,
    "aiv_mte2_ratio": 0.346, "aiv_mte2_time(ms)": 3.378,
    "aiv_mte3_ratio": 0.0, "aiv_mte3_time(ms)": 0.001,
    "aiv_scalar_ratio": 0.677, "aiv_scalar_time(ms)": 6.605,
    "aiv_time(ms)": 9.66, "aiv_total_cycles": 571879708.0,
    "aiv_ub_read_bw(GB/s)": 25.966, "aiv_ub_write_bw(GB/s)": 39.604,
    "aiv_vec_bank_cflt_ratio": 0.053, "aiv_vec_bankgroup_cflt_ratio": 0.058,
    "aiv_vec_ratio": 0.134, "aiv_vec_resc_cflt_ratio": 0.0,
    "aiv_vec_time(ms)": 1.31, "cube_utilization(%)": 0.0
  }
}
```

Key metrics include vector/scalar/MTE time ratios, unified buffer bandwidth, and cube utilization, enabling targeted optimization.

## H. Example Generated Kernel

The following shows a complete, correctly compiling Ascend C kernel for `Tanh` generated by EvoKernel. Each source file section is shown with a distinct background color. In actual verifier pipeline, the below artifact is parsed into respective files.

```
project_json_src = r'''
[
  {
    "op": "TanhCustom",
    "language": "cpp",
    "input_desc": [
      {
        "name": "x",
        "param_type": "required",
        "format": ["ND"],
        "type": ["float"]
```

```
        }
    ],
    "output_desc": [
      {
        "name": "y",
        "param_type": "required",
        "format": ["ND"],
        "type": ["float"]
      }
    ]
  }
]
'''
```

```
host_tiling_src = r"""
#include "register/tilingdata_base.h"

namespace optiling {
BEGIN_TILING_DATA_DEF(TilingData)
TILING_DATA_FIELD_DEF(uint32_t, totalLength);
TILING_DATA_FIELD_DEF(uint32_t, tileLength);
TILING_DATA_FIELD_DEF(uint32_t, blockDim);
END_TILING_DATA_DEF;

REGISTER_TILING_DATA_CLASS(TanhCustom, TilingData)
} // namespace optiling
"""
```

```
host_operator_src = r"""
#include "tanh_custom_tiling.h"
#include "register/op_def_registry.h"

namespace optiling {

static inline uint32_t AlignUp(uint32_t x, uint32_t a)
    { return (x + a - 1) / a * a; }
static inline uint32_t MinU32(uint32_t a, uint32_t b)
    { return a < b ? a : b; }

// Keep enough blocks for parallelism while ensuring decent work per block.
constexpr uint32_t MAX_BLOCK_DIM = 32;
// Larger tile to reduce DMA/queue overhead and improve effective bandwidth.
constexpr uint32_t DEFAULT_TILE_ELEMS = 8192;

static ge::graphStatus TilingFunc(gert::TilingContext *context)
{
    TilingData tiling;
    const uint32_t totalLength =
        context->GetInputShape(0)->GetOriginShape().GetShapeSize();
    const auto dtype = context->GetInputTensor(0)->GetDataType();
    if (dtype != ge::DT_FLOAT) {
        return ge::GRAPH_FAILED;
```

```
    }

    uint32_t blockDim = MAX_BLOCK_DIM;
    const uint32_t minElemsPerBlock = 262144;
    if (totalLength / blockDim < minElemsPerBlock) {
        blockDim = totalLength / minElemsPerBlock;
        if (blockDim < 1) blockDim = 1;
    }
    blockDim = MinU32(blockDim, MAX_BLOCK_DIM);

    uint32_t tile = DEFAULT_TILE_ELEMS;
    if (totalLength < tile) tile = totalLength;
    tile = AlignUp(tile, 8); // 32B alignment for fp32
    if (tile == 0) tile = 8;
    if (tile > totalLength) tile = totalLength;

    context->SetBlockDim(blockDim);
    tiling.set_totalLength(totalLength);
    tiling.set_tileLength(tile);
    tiling.set_blockDim(blockDim);

    tiling.SaveToBuffer(context->GetRawTilingData()->GetData(),
                        context->GetRawTilingData()->GetCapacity());
    context->GetRawTilingData()->SetDataSize(tiling.GetDataSize());

    size_t *ws = context->GetWorkspaceSizes(1);
    ws[0] = 0;
    return ge::GRAPH_SUCCESS;
}

} // namespace optiling

namespace ge {
static graphStatus InferShape(gert::InferShapeContext *context)
{
    *context->GetOutputShape(0) = *context->GetInputShape(0);
    return GRAPH_SUCCESS;
}

static graphStatus InferDataType(gert::InferDataTypeContext *context)
{
    context->SetOutputDataType(0, context->GetInputDataType(0));
    return ge::GRAPH_SUCCESS;
}
} // namespace ge

namespace ops {
class TanhCustom : public OpDef {
public:
    explicit TanhCustom(const char *name) : OpDef(name)
    {
        this->Input("x")
            .ParamType(REQUIRED)
            .DataType({ge::DT_FLOAT})
```

```
            .Format({ge::FORMAT_ND});
        this->Output("y")
            .ParamType(REQUIRED)
            .DataType({ge::DT_FLOAT})
            .Format({ge::FORMAT_ND});

        this->SetInferShape(ge::InferShape).SetInferDataType(ge::InferDataType);

        this->AICore()
            .SetTiling(optiling::TilingFunc)
            .AddConfig("ascend910b");
    }
};
OP_ADD(TanhCustom);
} // namespace ops
"""
```

```
kernel_src = r"""
#include "kernel_operator.h"

class KernelTanh {
public:
    __aicore__ inline KernelTanh() {}

    __aicore__ inline void Init(GM_ADDR x, GM_ADDR y, uint32_t totalLength,
        uint32_t tileLength, uint32_t blockDim)
    {
        this->totalLength = totalLength;
        this->tileLength = tileLength;
        this->blockDim = blockDim;

        const uint32_t bid = AscendC::GetBlockIdx();
        const uint32_t base = (totalLength / blockDim);
        const uint32_t rem = totalLength - base * blockDim;
        const uint32_t myLen = base + (bid < rem ? 1u : 0u);
        const uint32_t myOff = bid * base + (bid < rem ? bid : rem);

        startOffset = myOff;
        localLength = myLen;

        xGm.SetGlobalBuffer((__gm__ float*)x + startOffset, localLength);
        yGm.SetGlobalBuffer((__gm__ float*)y + startOffset, localLength);

        constexpr uint32_t PAD_BYTES = 256;

        pipe.InitBuffer(inQueueX, 2, tileLength * sizeof(float) + PAD_BYTES);
        pipe.InitBuffer(outQueueY, 2, tileLength * sizeof(float) + PAD_BYTES);

        // Tanh tmp UB, aligned to 256B. Keep conservative sizing.
        const uint32_t tmpBytes = ((tileLength * 8u + 255u) / 256u) * 256u;
        pipe.InitBuffer(tmpQueue, 1, tmpBytes);
    }
```

```cpp
    __aicore__ inline void Process()
    {
        if (localLength == 0) return;

        uint32_t offset = 0;

        uint32_t cur = (localLength > tileLength) ? tileLength : localLength;
        CopyIn(offset, cur);
        offset += cur;

        while (offset < localLength) {
            const uint32_t next = (offset + tileLength <= localLength) ?
                                   tileLength : (localLength - offset);

            CopyIn(offset, next);
            Compute(cur);
            CopyOut(offset - cur, cur);

            cur = next;
            offset += next;
        }

        Compute(cur);
        CopyOut(localLength - cur, cur);
    }

private:
    __aicore__ inline void CopyIn(uint32_t offset, uint32_t len)
    {
        AscendC::LocalTensor<float> xLocal = inQueueX.AllocTensor<float>();
        AscendC::DataCopy(xLocal, xGm[offset], len);
        inQueueX.EnQue(xLocal);
    }

    __aicore__ inline void Compute(uint32_t len)
    {
        AscendC::LocalTensor<float> xLocal = inQueueX.DeQue<float>();
        AscendC::LocalTensor<float> yLocal = outQueueY.AllocTensor<float>();

        AscendC::LocalTensor<uint8_t> tmp = tmpQueue.AllocTensor<uint8_t>();
        AscendC::Tanh<float>(yLocal, xLocal, tmp, len);
        tmpQueue.FreeTensor(tmp);

        outQueueY.EnQue<float>(yLocal);
        inQueueX.FreeTensor(xLocal);
    }

    __aicore__ inline void CopyOut(uint32_t offset, uint32_t len)
    {
        AscendC::LocalTensor<float> yLocal = outQueueY.DeQue<float>();
        AscendC::DataCopy(yGm[offset], yLocal, len);
        outQueueY.FreeTensor(yLocal);
    }
```

```cpp
private:
    AscendC::TPipe pipe;
    AscendC::TQue<AscendC::TPosition::VECIN, 2> inQueueX;
    AscendC::TQue<AscendC::TPosition::VECOUT, 2> outQueueY;
    AscendC::TQue<AscendC::TPosition::VECCALC, 1> tmpQueue;

    AscendC::GlobalTensor<float> xGm;
    AscendC::GlobalTensor<float> yGm;

    uint32_t totalLength = 0;
    uint32_t tileLength = 0;
    uint32_t blockDim = 1;

    uint32_t startOffset = 0;
    uint32_t localLength = 0;
};

extern "C" __global__ __aicore__ void tanh_custom(GM_ADDR x,
    GM_ADDR y, GM_ADDR workspace, GM_ADDR tiling)
{
    GET_TILING_DATA(tiling_data, tiling);
    KernelTanh op;
    op.Init(x, y, tiling_data.totalLength,
            tiling_data.tileLength, tiling_data.blockDim);
    op.Process();
}
"""
```

```cpp
python_bind_src = r"""
#include <torch/library.h>
#include <torch/extension.h>
#include "pytorch_npu_helper.hpp"

at::Tensor tanh_impl_npu(const at::Tensor& x) {
    auto y = at::empty_like(x);
    EXEC_NPU_CMD(aclnnTanhCustom, x, y);
    return y;
}

TORCH_LIBRARY_IMPL(myops, PrivateUse1, m) {
    m.impl("tanh_custom", &tanh_impl_npu);
}

PYBIND11_MODULE(TORCH_EXTENSION_NAME, m) {
    m.def("tanh_custom", &tanh_impl_npu, "tanh custom (NPU)");
}
"""
```

```python
model_src = r'''
import torch
import torch_npu
import custom_ops_lib
```

```
class ModelNew(torch.nn.Module):
    """
    Simple model that performs a Tanh activation using a custom Ascend C op.
    """
    def __init__(self):
        super(ModelNew, self).__init__()

    def forward(self, x: torch.Tensor) -> torch.Tensor:
        return custom_ops_lib.tanh_custom(x)
'''
```

