# OpenReview forum: "Towards Cold-Start Drafting and Continual Refining: A Value-Driven Memory Approach with Application to NPU Kernel Synthesis"
_ICML.cc/2026/Conference — ICML 2026 regular_

### Official Review · Reviewer_yejm · 2026-03-09

**Soundness:** 2
**Presentation:** 2
**Significance:** 2
**Originality:** 2
**Overall Recommendation:** 4
**Confidence:** 1

**Summary:**

The paper studies cold-start NPU kernel synthesis, where strong LLMs do well on CUDA but fail badly on Ascend C due to the lack of the domains-specific training data. It proposes EvoKernel, a two-stage system that first searches for a correct kernel and then refines it for lower latency, using a shared memory bank and value-based retrieval instead of model fine-tuning. Experiments on an Ascend version of KernelBench show large gains in correctness and good speedup from iterative refinement.

**Compliance With Llm Reviewing Policy:**

Affirmed.

**Final Justification:**

The rebuttal addressed my main questions reasonably well, especially by clarifying that the value update is better viewed as a bandit-style estimate rather than standard Q-learning, and by making the online adaptation setting more explicit. That said, I still have reservations about the limited originality of the overall approach, the imprecise use of terminology in the paper, and the gap between the claimed efficiency benefits and the results relative to Torch-NPU. I also think the paper would benefit from more careful proof reading and more precise calibration of its claims. Overall, I am slightly positive because the problem is important and the empirical gains in correctness are meaningful, but this is only a marginal weak accept from my side, and I would also understand a rejection decision.

**Key Questions For Authors:**

1. In Table 2, is the memory updated while evaluating the same benchmark set? If so, how should readers interpret these results compared with a strict held-out evaluation?

2. The value update in Eq. (4) looks closer to an EMA-style estimate of immediate utility than standard Q-learning, since there is no explicit state transition or bootstrapping term. Could the authors clarify whether this is intended as a simplified bandit-style value estimator, and how the state is actually defined in practice?

3. Figure 2 suggests that many generated kernels are still slower than Torch-NPU. Could the authors clarify how often the final refined kernels outperform Torch-NPU, and in what cases the method provides clear practical benefit?

**Limitations:**

The paper would benefit from a clearer discussion of its dependence on strong backbone models, the fairness of online memory accumulation during evaluation, and the gap between the refined kernels and strong vendor baselines such as Torch-NPU.

**Strengths And Weaknesses:**

## Strengths:
1. The motivation is clear and important. The large performance gap from CUDA to Ascend C in Table 1 makes the cold-start setting convincing.
2. The paper is easy to follow. The overall pipeline is clear, and most technical choices are reasonably justified.

## Weaknesses:
1. The Q-learning formulation is not fully convincing. The update in Eq. (4) does not involve state transition or long-term credit assignment, and looks closer to an EMA-style value estimate.
2. The performance is not as strong as the paper suggests. In Figure 2, many generated kernels are still slower than Torch-NPU, so the practical benefit on efficiency is less clear.
3. The originality is somewhat limited. The main contribution seems to be the application and integration of memory retrieval and iterative refinement, rather than a clearly new algorithmic idea.

---

> ### Author Rebuttal · Authors · 2026-03-30
>
> We thank the reviewer for the careful feedback and address the main concerns below.
>
> > **W1+Q2: Nature of the value update and state definition**
>
> We thank the Reviewer for highlighting a possible ambiguity here. Eq. (4) is a bandit-style EMA update rather than TD Q-learning, and this is by design: the verifier provides a complete outcome signal after each generation (correctness, latency), so the immediate reward fully evaluates the retrieval decision and there is no need to bootstrap from future steps. This keeps the system **simple and effective**. Extending to TD Q-learning is possible but would introduce additional complexity for uncertain gain.
>
> Concretely, the state $s_t$ captures (1) which operator is being solved, and (2) the current generation status (whether a feasible kernel exists and the best latency so far). This determines which reward function ($r_1$ or $r_2$) and value table ($Q_1$ or $Q_2$) apply, making each memory item's value state-dependent.
> Appendix A mentions "bandit-style", but the main text could be more explicit. We will clarify this and add a concrete state definition in Sec 3.1.
>
> > **W2+Q3: Latency gap with Torch-NPU and practical value**
>
> We appreciate this point. Since Ascend C kernel synthesis is a data-scarce domain far outside current models' pretraining distribution, and Torch-NPU is a mature vendor library backed by years of expert hand-tuning, a latency gap is expected. Replicating extensive vendor-level optimization from in-context examples alone remains challenging for current LLMs. With GPT-5.2, 12/166 correct operators match or outperform Torch-NPU, and we will report this explicitly.
>
> The main contribution of this work is not surpassing a vendor library but reliably producing correct kernels from a cold start and improving them through refinement: correctness improves from **4%** to **83%**, and the refining stage yields a **mean 12.96x speedup** over the first correct version. Per-category breakdown:
>
> |Category|#Kernels|Mean Speedup|
> |---|---|---|
> |Activation|14|7.48x|
> |Convolution|38|4.03x|
> |Fuse|76|15.91x|
> |Loss|9|71.40x|
> |Matmul|14|1.02x|
> |Normalization|6|1.35x|
> |Pooling|9|2.18x|
> |**Overall**|**166**|**12.96x**|
>
> In practice, the method is most valuable when a vendor-optimized kernel is absent (e.g., newly introduced or unsupported), or when developers need a working implementation for further manual tuning.
>
> > **W3: Novelty Concerns**
>
> We appreciate this concern and would like to clarify.
> Memory retrieval and iterative refinement are each very broad fields, and many widely recognized works compose them to address problems the parts alone could not, such as A-MEM [3], which combines memory retrieval with iterative memory refinement for LLM agents, and OpenHands [4], which combines code retrieval with iterative execution feedback for software development. Their contributions are recognized for how the composition is tailored to a specific problem, rather than discounted simply for lying at the intersection of two areas.
> EvoKernel follows the same principle, tailoring its composition to data-scarce kernel synthesis: retrieval is guided by stage-specific Q-values with different reward semantics across drafting (binary feasibility) and refining (relative latency), and the memory is seeded with external domain knowledge to mitigate the cold-start problem.
>
> > **Q1: Online memory update during evaluation**
>
> Yes, and this is by design. EvoKernel is evaluated as an online learning system: the goal is to measure what the agent achieves within a fixed interaction budget while accumulating experience, similar to AlphaEvolve [1] and EvolveR [2]. This should be interpreted as an online adaptation result, not a strict held-out generalization result.
> Importantly, the comparison is fair: all methods receive the same per-operator generation budget ($T=30$) and the same verifier feedback. The differences between methods, such as memory reuse and Q-value-guided retrieval, are part of the method design being evaluated rather than an asymmetry in experimental conditions.
>
> > **Limitation**
>
> Happy to discuss! The fairness of online memory accumulation and the gap with Torch-NPU are discussed in Q1 and W2+Q3 above, respectively. Regarding backbone dependence:
> The bottleneck is the backbone's **in-context learning capability**: Ascend C kernel synthesis is a data-scarce domain far outside pretraining distributions, so weaker models cannot build sufficiently high-quality experience on their own. The cross-backbone transfer experiment confirms this: when DeepSeek uses memory built by GPT-5.2, correctness jumps from **6%** to **58%**, showing that the framework and its accumulated experience remain effective. We expect this gap to narrow as open-source models continue to improve.
>
> [1] Novikov et al. (2025). AlphaEvolve. arXiv:2506.13131.
>
> [2] Wu et al. (2025). EvolveR. arXiv:2510.16079.
>
> [3] Xu, W., et al. (2025). A-MEM. NeurIPS 2025.
>
> [4] Wang, X., et al. (2025). OpenHands. ICLR 2025.

---

> > ### Author Rebuttal · Reviewer_yejm · 2026-04-02
> >
> > Thank you for the detailed rebuttal. The authors have addressed most of my concerns. I will therefore adjust my rating.

---

> > > ### Author Response · Authors · 2026-04-02
> > >
> > > We sincerely thank the reviewer for the constructive discussion and for re-evaluating our work. We will incorporate all promised revisions in the updated manuscript.

---

### Official Review · Reviewer_uKSF · 2026-03-10

**Soundness:** 4
**Presentation:** 3
**Significance:** 4
**Originality:** 3
**Overall Recommendation:** 5
**Confidence:** 4

**Summary:**

The paper introduces EvoKernel, an agentic framework designed to tackle the "cold-start" problem in synthesizing efficient kernels for Domain-Specific Architectures (DSAs), specifically focusing on Ascend C. Unlike CUDA, which benefits from massive pre-training data, NPUs suffer from a severe lack of training data, causing standard LLMs to fail in programming in such ecosystems with scarce data. EvoKernel formulates kernel synthesis as a Memory-based Markov Decision Process (M-MDP) and employs a reinforcement learning-backed, value-driven retrieval mechanism. The system operates in two distinct stages: "Drafting," which prioritizes to obtain initial feasible kernels using a binary reward, and "Continual Refining," which focuses on latency optimization using a continuous reward. By updating the Q-values of retrieved memory items (such as successful traces, API templates, and best practices) across tasks, the framework enables cross-task generalization and implicit curriculum learning. Empirical results demonstrate substantial improvements in compilation and correctness rates, along with significant latency speedups, particularly when utilizing highly capable frontier LLMs.

**Compliance With Llm Reviewing Policy:**

Affirmed.

**Final Justification:**

EvoKernel addresses a practically important and underexplored problem—cold-start kernel synthesis for data-scarce DSAs—with a technically sound dual-stage M-MDP formulation and comprehensive ablation studies. The rebuttal convincingly clarified the reward design rationale, provided failure mode analysis for weaker models, and demonstrated robustness to toolchain updates via the online Q-value mechanism. I raise my score accordingly.

**Key Questions For Authors:**

1. The authors present the two-stage process (Drafting vs. Refining) and the distinct reward functions purely formally, without explaining the reason behind the designs. Are these specific architectural choices (like Eq. 6) the result of a rigorous theoretical derivation? Or are they, conversely, the result of empirical engineering and trial-and-error which was then formalized into the M-MDP framework post-hoc? Could you please clarify the specific reason behind the reward function designs？
2. Given that Qwen3 and DeepSeek achieve 0% Acc on Level 2 tasks, could you provide insights into the specific failure modes? Could you please clarify whether this is due to the capability constraints of the specific model or issues inherent to the proposed framework?
3. If the AscendC toolchain receives a major update that deprecates certain functions, how robust is the value-driven memory? Does the framework require a complete reset of the Q-values, or can the negative rewards from compilation failures quickly down-weight obsolete best practices?

**Limitations:**

yes

**Strengths And Weaknesses:**

Strength:
- S1: The formulation of kernel synthesis as a dual-stage M-MDP is technically sound and highly appropriate for the domain. The scenario the paper faces is a highly critical bottleneck in modern AI systems and is valuable.
- S2: The experimental section and ablation studies are solid and comprehensive. The methodology is clear and logical. The discussion is well structured, providing a reasonable interpretation of the results.

Weakness:
- W1: The textual presentation is overly brief and lacks intuitive explanations for the key design choices in Sec3.3, Sec3.4.
- W2: The Acc and CR improvements are limited when using weaker models. The font sizes in several figures, Figure 2, 3, 7 for examples, are too small, impairing readability.

---

> ### Author Rebuttal · Authors · 2026-03-30
>
> We thank the Reviewer for the thoughtful and constructive feedback. We address each point below.
>
> > **W1+Q1: Reward design choices and their justification**
>
> We thank the Reviewer for raising this point. The design and formalization were developed reciprocally: the task structure determines the two-stage decomposition and reward form, while empirical feedback calibrates practical details. The two-stage decomposition reflects the task structure: correctness is a prerequisite for latency optimization, since a kernel that does not compile or produce correct outputs cannot be meaningfully optimized for speed. The Stage 1 reward is therefore binary, directly reflecting whether the candidate passes the feasibility gate. We chose not to use softer shaping signals such as partial compilation progress because they can reward code that appears promising but is still unusable end to end.
>
> The Stage 2 reward in Eq. 6 uses relative latency improvement rather than an additive difference because different operators can have very different runtime scales, so a ratio provides a more unified measure across tasks. We compare each candidate against the current best-so-far baseline $b_t$ because the objective in refining is to keep searching for a better implementation than the current best. The log transform makes gains and regressions more symmetric, and the $\tanh$ transformation keeps the raw reward bounded. The online normalization in Section 3.4 then handles the remaining non-stationarity: as $b_t$ improves over time and different tasks induce different reward scales, the PopArt-style running statistics $(\mu_2,\sigma_2)$ keep updates on a more comparable scale. In other words, Eq. 6 was not meant as a closed-form derivation, but as a simple reward aligned with the verifier structure of the problem and stable enough for online updating. More broadly, the M-MDP view provides a unified description of the two stages, and Appendix A analyzes boundedness of the value updates, stability of the normalization, and tracking/convergence under the assumptions.
>
> We will expand Sections 3.3 and 3.4 to make these design choices and their motivations more explicit.
>
> > **W2+Q2: Limited gains on weaker models, failure analysis, and figure readability**
>
> This is an important observation. Our evidence indicates that **this is primarily a model capability constraint, not an inherent limitation of the framework.** In our manual inspection of failed Level 2 cases for Qwen3 and DeepSeek V3.2, the dominant failure modes are compilation or deployment errors (e.g., hallucinated Ascend C APIs, invalid memory operations, or incomplete kernel project structure), incorrect outputs that do not match the PyTorch reference, and anti-hacking violations that bypass the required Ascend C implementation.
>
> Ascend C kernel synthesis is a data-scarce domain far outside current models' pretraining distribution, so the method relies heavily on the backbone's ability to learn from retrieved examples at inference time. This demand is especially acute for Level 2 operators, which require composing several Level 1 patterns into a new fused implementation. The cross-backbone transfer experiment further supports this diagnosis. When Qwen and DeepSeek are initialized with a GPT-5.2 memory bank instead of seeded memory alone, **performance improves substantially (DeepSeek: 6% to 58%; Qwen: 4% to 32%).** The framework and its accumulated experience remain effective; the bottleneck is that weaker backbones cannot build sufficiently high-quality experience on their own. Beyond that, we expect models with such capabilities will emerge rapidly.
>
> For readability, we will enlarge the x-axis labels and bottom legend in Fig. 2, split the four case-study subplots in Fig. 3 into a standalone figure, and increase the contrast of the category bubbles and enlarge the bottom legend in Fig. 7.
>
> > **Q3: Robustness to toolchain updates**
>
> We appreciate this practical concern. **No complete reset is generally required**, because the value update in Eq. 4 is designed to adapt online. If a toolchain update deprecates certain APIs, then whenever a generated operator fails verification for that reason, the retrieved memory items involved in that generation receive negative reward and their Q-values gradually decrease, so they need not remain competitive in retrieval and other possible optimization directions and starting points get explored. The agent thus **adapts incrementally across toolchain updates**. We will add a short discussion of this robustness consideration in the Discussion section.

---

> > ### Author Rebuttal · Reviewer_uKSF · 2026-04-04
> >
> > The authors addressed all my concerns satisfactorily.
> >
> > I will raise my score.

---

> > > ### Author Response · Authors · 2026-04-04
> > >
> > > We sincerely thank the reviewer for the constructive feedback and for revisiting our work. We greatly appreciate the updated evaluation and will reflect the discussed revisions in the revised manuscript.

---

### Official Review · Reviewer_FX39 · 2026-03-11

**Soundness:** 4
**Presentation:** 4
**Significance:** 2
**Originality:** 2
**Overall Recommendation:** 5
**Confidence:** 2

**Summary:**

the paper looks at an important problem: there's a data scarcity problem in newer domain-specific architectures, which means there's not a lot of expert demonstrations for models to learn from through pretraining or finetuning. this work proposes a framework that allows a language model (LM) to explore and store its experiences somewhere for future use. The authors built an NPU variant of kernelbench and evaluated the models on it. the frontier model's correctness performance improved from 11% to 83% and achieved a 3.6x speedup.

**Compliance With Llm Reviewing Policy:**

Affirmed.

**Final Justification:**

the rebuttal addressed my main concerns.

**Key Questions For Authors:**

Could this approach generalize to new domains? (like a different DSL)

**Limitations:**

more explicit discussion on limitations is needed. for instance, memory-augmented approaches only work on tasks that are very similar. so you need to construct such a memory bank for each specialized task.

**Strengths And Weaknesses:**

- Soundness
  - strengths: the submission is technically sound. the experiments are thoughtfully designed. the use of multi-gate verification is also reasonable.
  - weaknesses: it would be good to see more ablations that can isolate the impact of value function design from memory bank quality.

- Presentation
  - strength: well-written, very clear. related work section is thorough

- significance
  - strength: the paper addresses an important problem. it shows that a simple inference-time approach could achieve significant performance gains
  - weaknesses: requires a strong backbone model, requires tasks to have shared structures in order for memory augmentation to be useful.

- originality
  - strengths: the two stage (feasibility and latency optimization) decomposition is novel design choice, novel evaluation data
  - weaknesses: the idea of memeory RL is not new, this type of self-improving agent had been proposed in prior work like reflexion.

---

> ### Author Rebuttal · Authors · 2026-03-30
>
> We thank the Reviewer for the positive assessment and constructive feedback.
>
> >**W1: Isolating the Effect of Value Function Design**
>
> This is a valuable suggestion, because the Value-Driven and Heuristic-Driven experiment in Sec. 4.4.1 accumulates different memory over time. To address this more directly, we use the memory accumulated in the L1$\rightarrow$L2 generalization experiment, since the original Value-Driven vs. Heuristic-Driven comparison is also conducted under that protocol. We select five memory snapshots from the value-driven run at different iterations and compare the original retrieval policy against the same memory with Q-values removed.
>
> | Iteration |CR w/ Q (%) |CR w/o Q (%)| Acc w/ Q (%) | Acc w/o Q (%) |
> |---|---:|---:|---:|---:|
> | 5  | 25 | 33 | 4  | 9  |
> | 10 | 48 | 55 | 14 | 14 |
> | 15 | 61 | 63 | 26 | 28 |
> | 20 | 64 | 63 | 31 | 22 |
> | 25 | 69 | 71 | 39 | 30 |
> | 30 | 68 | 65 | 40 | 31 |
>
> The main pattern is that compilation changes only modestly, while correctness diverges more clearly in later iterations: at iter 20/25/30, correctness drops from 31/39/40% to 22/30/31% when Q-values are removed. This trend is also consistent with the original discussion in Sec. 4.4.1: heuristic retrieval can be competitive during early bootstrapping, while learned Q-values become more useful later. We will include this ablation in Sec. 4.4 to make the role of the value function more explicit.
>
> >**W2: Dependence on a Strong Backbone and Shared Task Structure**
>
> We appreciate the Reviewer raising these two concerns together. We agree that backbone capability matters, but do not view this as specific to EvoKernel. The core issue is that Ascend C kernel synthesis is a **data-scarce domain** far outside current models' pretraining distribution, so the method depends on the backbone being able to learn effectively from retrieved examples at inference time, which places a high demand on **in-context learning capability**. This is reflected in Table 2, where weaker models remain at 0% L2 accuracy. However, weaker models can still improve substantially when equipped with memory built by GPT-5.2: for DeepSeek, compilation improves from **26%** to **80%** and correctness from **6%** to **58%**. With fast iterations of foundation models, we expect models with such capabilities will emerge rapidly.
>
> On the second point, memory augmentation does not require tasks to share the same implementation structure as a hard condition. In our current system, task-level structural similarity is mainly a practical retrieval signal for surfacing semantically related prior operator experience, rather than the essence of what makes transfer work. What ultimately matters is whether the current operator can retrieve prior experience that is close in **semantics or functional requirements**. As a result, two tasks may differ at the full-program level and still benefit from transfer if they contain operator subproblems with similar needs. We will clarify in Sec. 3.2 that task-structure similarity is an effective implementation mechanism in our current system, not the sole prerequisite for memory augmentation to help.
>
> >**W3: Concerns on Novelty**
>
>
> We thank the Reviewer for raising this point. EvoKernel is not a per-task self-improvement loop with task-local memory. Instead, it maintains a **single shared memory bank across all tasks** (Sec. 3.2), and every interaction updates that shared bank through Eq. 1. Thus, experience gained from one task can later be reused for another, rather than remaining confined to a single task episode. This cross-task reuse is central to the performance gain: in Sec. 4.4.2, restricting memory to a single kernel (the Refinement baseline) yields only 3% L2 accuracy, whereas the shared-memory version reaches 76%. This is the main difference from Reflexion-style within-episode self-improvement, and we will make it more explicit in Related Work and Sec. 3.2.
>
> >**Q1: Generalization to New Domains**
>
> Yes. Since submission, we have extended EvoKernel from Ascend C to **CUDA** without changing the core algorithm; only the prompt template and verifier backend are replaced.
>
> |Setting|DSL|#Ops|Correct (%)|Faster than PyTorch (%)|
> |---|---|---|---|---|
> |KernelBench|CUDA|250|100.0|68.0|
> |Attention Set|CUDA|70|97.1|72.1|
>
> These results show that the framework is **not tied to Ascend C** and transfers across DSLs and operator families with minimal re-instantiation. In other words, the domain-specific parts are the seeded knowledge and verifier, not the core M-MDP/value-driven retrieval mechanism. We will add this cross-DSL result to the revision.

---

> > ### Author Rebuttal · Reviewer_FX39 · 2026-04-03
> >
> > the authors addressed my concerns and added results on new domain. I will raise my score

---

> > > ### Author Response · Authors · 2026-04-03
> > >
> > > We thank the reviewer for raising the score and for the constructive discussion during the rebuttal period. All improvements discussed will be incorporated into the revised version.

---

### Official Review · Reviewer_oK6P · 2026-03-15

**Soundness:** 3
**Presentation:** 2
**Significance:** 2
**Originality:** 2
**Overall Recommendation:** 3
**Confidence:** 3

**Summary:**

This paper proposes EvoKernel, a kernel-generation agent that enables self-evolving memory through a value-based memory retrieval mechanism. By integrating memory into a unified generation-and-optimization workflow, EvoKernel iteratively improves kernel quality. On the AscendC version of KernelBench, it outperforms advanced commercial models and frameworks such as GPT-5.2 and Codex.

**Compliance With Llm Reviewing Policy:**

Affirmed.

**Key Questions For Authors:**

1. Novelty Compared to Prior Memory-Based Methods.  Compared with MemRL and Memento, does EvoKernel introduce any task-specific innovations in memory design or utilization for kernel generation? More specifically, is this work primarily an engineering adaptation of an existing agent framework, or does it propose fundamentally new mechanisms?

2. Form and Evolution of Memory. During the iterative generation and optimization process, what is the structure of the gradually accumulated memory? Providing concrete examples would greatly improve clarity.

3. Search Strategy in Optimization. Does the optimization process incorporate any search strategies such as Monte Carlo Tree Search, beam search, or other structured exploration methods?
    - If multiple feasible solutions are generated, how does the agent system select which candidate to optimize further?
    - If a search strategy is used, please provide detailed descriptions.
    - If not, please explain why such strategies were not adopted and how exploration–exploitation trade-offs are handled.

**Limitations:**

Yes

**Strengths And Weaknesses:**

Strengths:

1. The paper presents a unified workflow that integrates both generation and optimization for the AscendC kernel synthesis.
2. The proposed method achieves strong empirical performance on the AscendC version of KernelBench.
3. The experiments include analysis of memory generalization, providing insight into the transferability of stored knowledge.

Weakness：

1. The methodological novelty is limited. Value-based memory retrieval has been explored in prior work. The paper should more clearly articulate how EvoKernel differs from existing approaches such as MemRL and Memento.
2. The experimental evaluation lacks comparisons with other state-of-the-art methods specifically designed for kernel generation or code optimization. The refinement and Codex baselines are not tailored to this domain, which weakens the fairness and strength of the comparisons.

---

> ### Author Rebuttal · Authors · 2026-03-30
>
> We thank the Reviewer for the detailed and constructive feedback. We address each point below.
>
> > **W1&Q1: Novelty Comparison**
>
> We thank the Reviewer for raising this question. MemRL and Memento establish the general M-MDP paradigm for memory-augmented agents, but do not target the cold-start kernel-synthesis setting. Our claim is not that value-based memory retrieval itself is new. Rather, EvoKernel adapts this paradigm to this setting with several task-specific design choices.
>
> 1. EvoKernel uses a **hybrid memory with pre-seeded external knowledge**. MemRL and Memento rely entirely on self-bootstrapped memory. In a cold-start setting where the target DSL (Ascend C) is absent from pre-training data, **pure self-bootstrapping cannot get off the ground**. EvoKernel therefore initializes memory with domain knowledge such as API templates, and architecture constraints alongside self-generated experiences. The learned Q-values then determine how useful each item is in practice.
>
> 2. EvoKernel retains a **dual-stage design specific to the kernel-synthesis objective**. Because correctness must precede latency optimization, the two stages retrieve differently. Stage 1 resembles prior methods: dense similarity forms a candidate pool $\mathcal{C}(x)$ and $Q_1$ selects context to bootstrap a correct kernel. Stage 2 selects an optimization start point from $\mathcal{P}(x)$ via $Q_2$ and retrieves context around it. The correct kernel from Stage 1 directly seeds Stage 2, and **Q-values evolve across both stages and tasks**.
>
> We note that these distinctions are discussed in Sec. 2 and operationalized in Secs. 3.1–3.4; we will make this comparison more explicit in the revision.
>
> >**W2: Experimental Comparison**
>
> Thanks for that insight! We view AscendKernelGen [1] as concurrent work whose public code became available after our submission. We ran KernelGen-LM-32B-RL on 40 randomly sampled KernelBench operators (20 level-1, 20 level-2) with 30 candidates/operator, matching our 30-iteration budget.
>
> |Method|CR|Acc|L1 CR|L2 Acc|
> |---|---|---|---|---|
> |KernelGen-LM-32B-RL|67.5%|5.0%|10.0%|0.0%|
>
> Thus, **the baseline rarely reaches end-to-end functional correctness** equivalent to ours under a matched budget, primarily rooting from them having a different yet easier testbed than ours. We will add this concurrent-work comparison and clarify the setup in the revision.
>
> >**Q2: Form and Evolution of Memory**
>
> We are happy to clarify. As described in Sec. 3.2, the memory contains four item types: API templates, success/failure experiences, generation traces, and refinement best practices. API templates and best practices are pre-seeded as $\mathcal{M}_0$, while the other two accumulate online.
> The memory evolves via two concurrent mechanisms.
>
> 1. item accumulation: after each generation attempt, the produced code, experience, and verifier feedback are stored into $\mathcal{M}$.
>
> 2. value refinement: each memory item carries a Q-value that evolves with retrieval outcomes. **Items that contribute to successful generations have their Q-values increased and are retrieved more frequently; those that do not have their Q-values decreased and are gradually filtered out.** Figure 7 illustrates this for `36_RMSNorm`: early iterations retrieve from `22_Tanh` via dense similarity and Q-value selection, but as Q-values update across iterations, retrieval shifts to normalization-related operators — driven by Q-value evolution, not similarity.
>
> >**Q3: Search Strategy in Optimization**
>
> Our optimization stage selects start points via **Q-value-based epsilon-greedy** over a dynamically maintained pool of verified kernels, rather than tree-structured search.
> After Stage 1 finds the first correct kernel, Stage 2 maintains a candidate pool $\mathcal{P}(x)$ of feasible kernels. At each iteration, the agent selects a start point from $\mathcal{P}(x)$ based on $Q_2$, retrieves relevant context conditioned on that start point, generates a refined kernel, adds it back to $\mathcal{P}(x)$ if verified correct, and updates the $Q_2$ of the selected start point based on the observed latency reward. So when multiple feasible solutions exist, the system does not simply continue from the fastest kernel — $Q_2$ estimates each candidate's potential for further improvement, not its current performance.
>
> The exploration-exploitation trade-off is handled at two levels.
>
> 1. epsilon-greedy selection:
> with probability $\epsilon$ the agent explores uniformly, and otherwise it exploits the highest-$Q_2$ candidate.
>
> 2. non-stationary Q-values:
> as the current best changes, reward shifts accordingly, so a candidate that was **promising earlier may become less attractive, and vice versa, causing exploration on new candidates**. Thus exploration and exploitation persist throughout the optimization process.
>
> [1] Cao, X., et al. (2026). AscendKernelGen: A Systematic Study of LLM-Based Kernel Generation for Neural Processing Units. arXiv:2601.07160.

---

> > ### Author Rebuttal · Reviewer_oK6P · 2026-04-04
> >
> > Thanks to the rebuttal from the authors, and I decide to increase my score.

---

> > > ### Author Response · Authors · 2026-04-05
> > >
> > > Thank you for your positive response to our rebuttal and for acknowledging that you will raise your score accordingly. We truly appreciate it. All improvements discussed will be incorporated into the revised manuscript.
> > >
> > > Besides, we are currently seeing the original score of 3 on the review form still unchanged. Could you please update the rating field so that the change is reflected? Thank you again!

---

### Decision · Program_Chairs · 2026-04-30

**Decision:**

Accept (regular)

**Comment:**

This paper introduces EvoKernel, a framework that formulates NPU kernel synthesis as a memory-based MDP with a two-stage value-driven retrieval mechanism.

The results are striking, improving frontier-model correctness from 11.0% to 83.0% on an NPU variant of KernelBench, alongside a 3.60× median speedup. The reviewing panel (2 Accepts, 1 Weak Accept, 1 Weak Reject) agreed that the paper addresses a critical, under-explored problem: cold-start kernel synthesis on data-scarce DSAs, where general-purpose LLMs typically struggle.

The primary shared concern was that the method is largely an integration of existing techniques (memory retrieval + iterative refinement) rather than a fundamentally new mechanism. Furthermore, the refined kernels still fall short of vendor-tuned Torch-NPU baselines. However, the authors substantively addressed these critiques in the rebuttal with targeted ablations and improved presentation.

 Aligning with the majority of reviewers, I find that the coherent methodological design and the massive empirical gains in an underserved hardware ecosystem outweigh the incremental nature of the algorithmic novelty. Therefore, I recommend Weak Accept.